



# Cryoconite as an efficient monitor for the deposition of radioactive fallout in glacial environments

Giovanni Baccolo[1,2], Edyta Łokas[3], Paweł Gaca[4], Dario Massabò[5,6], Roberto Ambrosini[7], Roberto S. Azzoni[7], Caroline Clason[8], Biagio Di Mauro[1], Andrea Franzetti[1], Massimiliano Nastasi[1,9], Michele Prata[10], Paolo Prati[5,6], Ezio Previtali[1,9], Barbara Delmonte[1], Valter Maggi[1,2]

[1]     Environmental and Earth Sciences Department, University of Milano-Bicocca, Milano, 20126, Italy.
[2]     INFN section of Milano-Bicocca, Milano, 20126, Italy.
[3]     Department of Nuclear Physical Chemistry, Institute of Nuclear Physics Polish Academy of Sciences, Kraków, 31-342, Poland.
[4]     Ocean and Earth Science, University of Southampton, National Oceanography Centre, Southampton, SO14 3ZH UK.
[5]     Physics Department, University of Genoa, Genoa, 16146, Italy.
[6]     INFN section of Genoa, Genoa, 16146, Italy.
[7]     Department of Environmental Science and Policy, University of Milan, Milano, 20133, Italy.
[8]     School of Geography, Earth and Environmental Sciences, University of Plymouth, Plymouth, PL48AA, UK.
[9]     Physics Department, University of Milano-Bicocca, Milano, 20126, Italy.
[10]     Laboratory of Applied Nuclear Energy, University of Pavia, Pavia, 27100, Italy.

*Correspondence to*: G. Baccolo (giovanni.baccolo@unimib.it)

**Abstract.** Cryoconite is extremely rich in natural and artificial radionuclides, but a comprehensive discussion about its ability to accumulate radioactivity is lacking. A characterization of cryoconite from two Alpine glaciers is presented and discussed. Results confirm that cryoconite is among the most radioactive environmental matrices, with activity concentrations exceeding 10,000 Bq kg$^{-1}$ for single radionuclides. Atomic and activity ratios of Pu and Cs radioactive isotopes reveal that the artificial radioactivity of Alpine cryoconite is mostly related to the stratospheric fallout from nuclear weapon tests and to the 1986 Chernobyl accidents. The signature of cryoconite radioactivity is thus influenced by both local and more widespread events. The extreme accumulation of radioactivity in cryoconite can be explained only considering the glacial environment as a whole, and particularly the interaction between ice, meltwater, cryoconite and atmospheric deposition. Cryoconite is an ideal monitor to investigate the deposition and occurrence of natural and artificial radioactive species in glacial environment.

## 1 Introduction

Radioecological research is primarily focused on Earth surface environments, where continuous atmospheric deposition of fallout radionuclides (FRN), both natural and artificial, are accumulated. The most common FRNs are cosmogenic nuclides, $^{222}$Rn progeny and artificial products. The latter of which have been released into the environment since the second half of



the twentieth century, as a consequence of nuclear test explosions and accidents. Hundreds of thousands of PBq were spread

in the high troposphere and stratosphere from the 1940s to the 1960s, allowing for global dispersion and contamination at the Earth surface. The extent and impact of FRN deposition on the Earth surface is monitored through the analysis of different environmental matrices, which are used to reconstruct FRN deposition (Steinhauser et al., 2013) and understand their environmental mobility and distribution (Avery et al., 1996; Yasunari et al., 2011).

Among the matrices used in the study of FRNs, those that receive the greatest attention share common features: their

composition is directly influenced by airborne deposited impurities; the contribution from environmental compartments other than atmosphere is limited; they are widespread, accessible and preferably easy to sample. Given these attributes, lichens, mosses and peat are commonly used to study the distribution of FRNs and establish depositional inventories (Nifontova, 1995; Kirchner and Daillant, 2002).

In recent years cryoconite began drawing attention in the field of radioactive environmental monitoring as an alternative

environmental matrix. Cryoconite is the dark, incoherent sediment that is found on the surface of glaciers worldwide (Takeuchi et al., 2001). It forms exclusively at the ice-atmosphere interface and in presence of abundant meltwater. It can be found as a dispersed material or as a deposit accumulated on the bottom of characteristic water-filled holes melted into ice (cryoconite holes, see **Fig. 1**). Cryoconite forms out of the interaction between the mineral particles present on the ice surface (both allochthonous and autochthonous), and the complex microbial communities that develop on the surface of

glaciers (Cook et al., 2015). Among the microbes that are present on glaciers, cyanobacteria play a major structural role. During the ablation season, when liquid water is available at the surface of glaciers, cyanobacteria develop films and filaments that promote the formation of aggregates composed of mineral sediments and organic matter, resulting in cryoconite (Takeuchi et al., 2001). The composition of cryoconite is dominated by a mineral component accounting for 85-95 % of its mass, whereas the remnant fraction is comprised of both living and dead organic matter and is responsible for its

dark colour (Cook et al., 2015). The formation of cryoconite holes is attributable to the dark colour, and thus low albedo, of cryoconite, which enhances the absorption of incoming solar radiation and locally increases ice melting to foster the development of holes in the ice surface. Due to its contribution to ice surface melting, it's diverse composition, and the role in biodiversity, cryoconite has been studied by a range of disciplines, including glaciology, microbiology, biogeochemistry and ecology. More recently cryoconite has been the subject of renewed interest due to its ability to accumulate specific

substances, including anthropogenic contaminants.

To the best of our knowledge, the first evidence of the accumulation of radionuclides in cryoconite was reported in 1996 by Tomadin et al. who found high levels of anthropogenic radioactivity in cryoconite samples from the European Alps. The following year Meese et al. analysed cryoconite formed on the surface of multi-year Arctic sea ice, measuring high radioactivity values. Since these early findings, no other studies (to our knowledge) have focused on the radioactivity of

cryoconite until the presentation of an extensive characterization of samples from an Austrian glacier in 2009 (Tieber et al.,



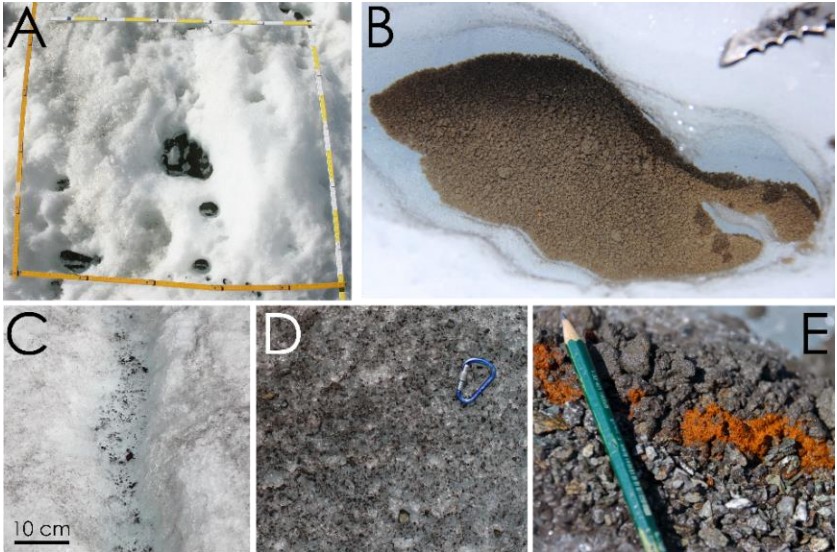

**Fig. 1 Cryoconite sampled on the Morteratsch and Forni glaciers. Panels A and B: two cryoconite holes sampled on the surface of the Forni glacier (A) and the Morteratsch glacier (B). Panel C: a snapshot from the Morteratsch glacier at the beginning of the melt season (early July). The ice surface has been recently exposed to the atmosphere after snow melting, and cryoconite is preferentially accumulating within the early meltwater channels. Panel D: a second snapshot from the Morteratsch glacier at the end of the melt season (late September). The ice surface has experienced three months of severe melting and is largely covered by cryoconite by this time. Panel E: this cryoconite deposit was located few meters downstream from a melting Spring snow patch rich in Saharan dust. Cryoconite acted as a filter and separated the dust particles from meltwater, retaining and accumulating them.**

2009). That study has showed that cryoconite is contaminated not only by $^{137}$Cs, a common artificial radionuclide spread into the environment, but also by several other species, both artificial and natural in origin. The reported activity levels in this study are extremely high, in some cases exceeding 10,000 Bq kg$^{-1}$, comparable to soil samples from nuclear incident and explosion sites (Abella et al., 2019; Steinhauser et al., 2014). Subsequent studies, carried out in different regions of global cryosphere, have corroborated the ability of cryoconite to efficiently accumulate radionuclides in the European Alps, the Caucasus, and the Svalbard (Baccolo et al., 2017; Łokas et al., 2016, 2018), with radioecological consequences concerning the presence of FRNs that are not limited to glaciers, but extend to the pro-glacial areas (Łokas et al., 2017). In addition to FRNs, cryoconite has also been shown to accumulate other anthropogenic contaminants including heavy metals (Baccolo et al., 2017; Łokas et al., 2016; Singh et al., 2017), artificial organic compounds (Ferrario et al., 2017; Weiland-Bräuer et al., 2017), and even microplastics (Ambrosini et al., 2019). However, among the species found in cryoconite, radionuclides show by far the highest concentration with respect to other environmental matrices. The considerable activity concentrations of artificial FRNs found in cryoconite, has allowed for the application of cutting-edge radiological techniques, offering important insight to the sources and distribution of radioactivity deposited within the glaciated regions of the Earth (Łokas et al., 2018, 2019), and making novel contributions to the emerging field of environmental nuclear forensics (Steinhauser, 2019).



Even though the link between cryoconite and environmental radioactivity is now indisputable, several important questions remain. Firstly, is not clear why and how cryoconite accumulates radioactivity and to what extent this accumulation is related to processes specific to glacial environments. Previous studies (Osburn Jr., 1963; Pourcelot et al., 2003) have focused on the role of snowmelt in accumulating radionuclides in residual snow patches during summer, suggesting that nival and melt processes could encourage local accumulation of radioactivity in a similar fashion to what has been observed in cryoconite on glaciers. Secondly, it is not fully understood where this radioactivity comes from and if its signature is local, regional or even more widespread.

This paper aims to present cryoconite as a promising tool for radioecological monitoring in high latitude and high-altitude areas and shed light on some of the open issues related to the themes explored above. Data concerning cryoconite from two European Alpine glaciers are presented and compared in detail to data from previous studies on both cryoconite and other environmental matrices used for radioactive monitoring.

## 2 Study site and sampling strategy

The Morteratsch and Forni Glaciers are located in the central sector of the European Alps and are situated ~50 km apart (Fig. 2). Both glaciers are among the biggest and most studied in the Alps. The Morteratsch is a Swiss glacier and the largest in the Bernina range, spanning an altitudinal range of roughly 2000 m and with a terminus located at 2100 m a.s.l.. It has an area of ~7.5 km$^2$, but until few years ago the value exceeded 15 km$^2$ due to its connection with a tributary glacier that has since detached. A similar setting characterizes the Forni Glacier, the biggest glacier of the Ortles-Cevedale range in Italy. Forni is a north-facing valley glacier presenting a glacial tongue that was, until recently, fed by three accumulation basins. Owing to retreat of the glacier, the connections between the basins have become weaker and the glacier is now fragmented in two distinct parts (Azzoni et al., 2017). The Forni glacier ranges between 2500 to 3750 m a.s.l. and is characterized by a surface area of ~10 km$^2$ considering both ice bodies. From a climatic perspective, the glaciers are similar; they are both characterized by a continental climate and in the last years experienced a darkening of their ablation zones because of the accumulation of impurities on the ice surface and the progressive emergence of detritus from medial and lateral moraines (Di Mauro et al., 2017; Fugazza et al., 2019). These processes are extremely favourable for the formation of cryoconite, and these two glaciers are two of the most studied in the Alps in terms of cryoconite and its components (Ambrosini et al., 2019; Baccolo et al., 2017; Di Mauro et al., 2017; Fugazza et al., 2019; Pittino et al., 2019).

Samples considered in this work were collected in summer 2015 (July and September) and 2016 (July) from the ablation zones of both glaciers (Fig. 2). Each sample represents a distinct cryoconite hole. The sampling was carried out using clean



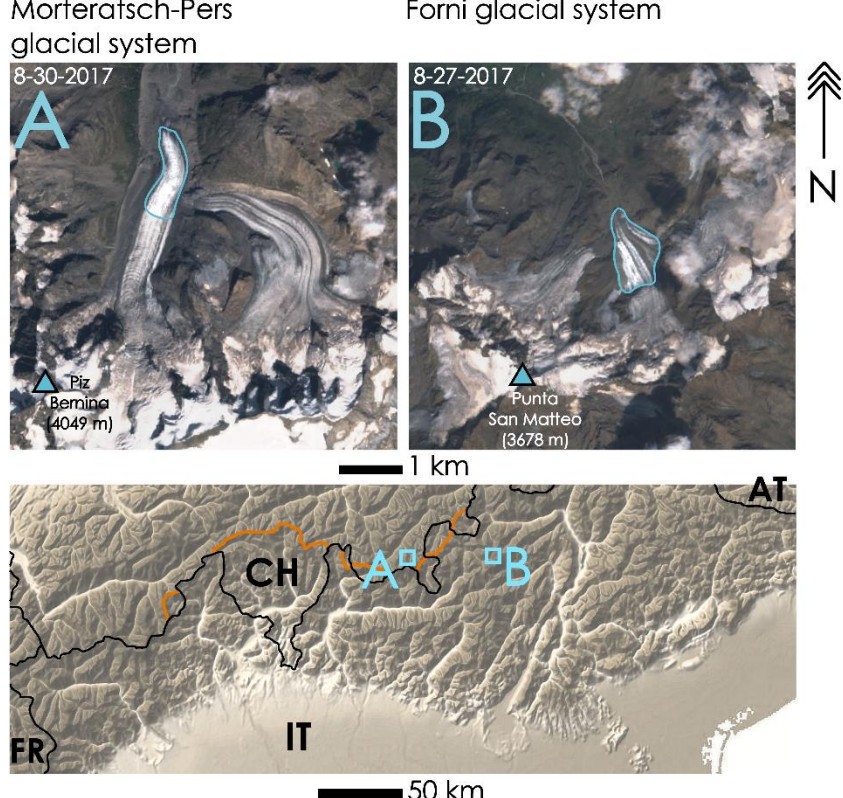

**Fig. 2 The geographic setting of the present work. Satellite images (ESA Sentinel-2, dates are reported) of the two glaciers considered in this study (panels A and B). The blue triangles highlight the highest peak point within each of the two catchments, and the blue lines define the ablation areas where cryoconite samples were collected. Panel C: a wider view of the central sector of the European Alps. The black line represents national borders, while the orange line, when not coincident with the black one, represents the Northern-Southern Alpine watershed dividing line.**

pipettes or spoons and samples were kept at 0°C during the field campaign and successively stored at -20°C in the EUROCOLD laboratory of the University Milano-Bicocca, until preparation for the geochemical analyses. Twelve samples have been gathered on the Morteratsch Glacier and ten on the Forni Glacier. Where possible, multiple analyses have been carried out on the same samples, but this has not been always possible, due to limited sample availability. Part of the dataset emerging from the fieldwork on the Morteratsch glacier (data concerning gamma spectroscopy) has already been published (Baccolo et al., 2017), however the remaining samples and results are presented here for the first time.

## 3 Materials and Methods

### 3.1 Radioactivity measurements

The activity of natural and artificial radionuclides present in cryoconite, was measured using a number of techniques. $^{137}$Cs, $^{241}$Am, $^{207}$Bi, $^{40}$K and $^{238}$U and $^{232}$Th decay chain nuclides were analysed through γ-spectroscopy about six months after





sampling, full details are found in the supplementary material. Aliquots dedicated to γ-spectroscopy consisted in ~1 g of dry material (dried until constant weight at 50°C, 2 mm sieved) sealed in polyethylene vials. The acquisition of the γ-spectra took place at least two weeks after the sealing, to allow the secular equilibrium between $^{222}$Rn and its progenies to be attained. Each sample was counted for about one week using a customized high purity germanium well detector (Ortec).

Details about the instrument, calibration and analytical performances are presented elsewhere (Baccolo et al., 2017). Aliquots dedicated to Pu analyses (~1 g of dry material) were ashed at 600° C to remove organic matter. The ash was dissolved with mineral acids and the resultant liquid samples underwent radiochemical separation and concentration of Pu isotopes. The procedure is extensively described elsewhere by Łokas and colleagues (2016). Activities of $^{239+240}$Pu and $^{238}$Pu were determined through α-spectrometry after Pu isotopes co-precipitation with NdF$_3$, using Canberra 7401 and Ortec Alpha

Duo spectrometers. After further radiochemical purification procedures, the $^{240}$Pu/$^{239}$Pu atomic ratio was measured through MC-ICP-MS (Thermo Fisher Scientific Neptune spectrometer), in accordance with Łokas et alii (2018). $^{238}$U and $^{232}$Th activities were not directly measured but were estimated considering the total content of U and Th.

### 3.2 Instrumental neutron activation analysis

The Th and U composition of cryoconite samples was assessed through instrumental neutron activation. Samples were

irradiated at the LENA laboratory, where a TRIGA Mark II nuclear reactor is available for research. The irradiation lasted for six hours under a thermal neutron flux of $2.4 \pm 0.2 \cdot 10^{12}$ neutron s$^{-1}$ cm$^{-2}$. To determine the concentration of Th and U in the samples, the following nuclear reactions and γ-emissions were exploited: $^{232}$Th (n,γ) $^{233}$Th → $^{233}$Pa (analyzed emissions: 300.3 and 312.2 keV) and $^{238}$U (n,γ) $^{239}$U → $^{239}$Np (analyzed emissions: 228.2 and 277.6 keV). Each irradiated sample was counted for six hours a few days after the irradiation, using the same well detector applied for γ-spectrometry. The

quantification of the elemental concentrations was carried out in accordance to a relative method, comparing irradiated samples with irradiated reference materials. Full details are given in a previous publication (Baccolo et al., 2017).

### 3.3 Carbonaceous content

A thermo-optical analyzer (Sunset Lab Inc. analyzer) was used for the determination of organic and elemental carbon content (OC and EC respectively), following the protocol adopted in Baccolo et al. (2017). Cryoconite samples were

suspended on clean quartz fiber filters and analyzed. The mass concentration of OC and EC was obtained combining the information relative to filter superficial concentrations and the mass of cryoconite deposited on the latter, determined using an analytical microbalance (precision 1 µg) which was operated inside an air-conditioned room (T = 20 ± 1 °C; relative humidity = 50 ± 5 %).

### 3.4 Statistics

To evaluate the degree of correlation between the variables and the samples, two multivariate statistical tools were applied. Multidimensional scaling (MDS) was used to appreciate the degree of correlation between the radionuclides, for an overview



of the method and of the required calculation please see the work from Diaconis et al. (2008). MDS was applied to a similarity metric derived from the correlation matrix (Pearson correlation) of the original data, following Eq. 1, where the distance $d$ between two variables $v_1$ and $v_2$ is obtained considering their Pearson correlation coefficient ($r$). In accordance to

this metric distance (van Dongen and Enright, 2012), two perfectly correlated (or anticorrelated) variables ($r = \pm 1$) have a null distance, while two uncorrelated variables ($r = 0$) have a maximum distance equal to 1.

$$d(v_1, v_2) = \sqrt{1 - r(v_1, v_2)^2}$$

**Eq. 1**

The correlation between samples and the differences between the two glaciers were evaluated applying the principal
component method to standardized data. The first two components (which explain 65 % of the total variance) were taken into consideration.

## 4 Results and discussion

### 4.1 Cryoconite natural radioactivity

The ability of cryoconite to accumulate radioactivity is now recognized within a number of previous research efforts and
multiple locations around the world (Baccolo et al., 2017; Łokas et al., 2016, 2018; Tieber et al., 2009). Results from the Forni and Morteratsch glacier samples further support this process of accumulation, with anomalously high activities found for the majority of the analysed radionuclides. The common factor shared by the enriched radionuclides is their primary source. Only FRNs, whose distribution is related to atmospheric transport, are actually accumulated in cryoconite, not the lithogenic ones. This feature can be observed in Fig. 3, where the activity of lithogenic radionuclides is presented. A
substantial secular equilibrium characterizes the nuclide belonging to the $^{238}$U and $^{232}$Th decay chains, except for $^{210}$Pb ($t_{1/2} = $ 22.3 yr). $^{210}$Pb presents an excess with respect to the other $^{238}$U-related nuclides. Excluding this nuclide, the average $^{238}$U and $^{232}$Th chain activities are $70 \pm 15$ and $52 \pm 8$ Bq kg$^{-1}$ respectively for the $^{238}$U chain (Morteratsch and Forni samples, $\pm$ standard deviation) and $50 \pm 10$ and $55 \pm 10$ Bq kg$^{-1}$ for the $^{232}$Th chain. These values, as seen in Fig. 3, are comparable to the average $^{238}$U and $^{232}$Th radioactivity of upper continental crust (UCC) reference (Rudnick and Gao, 2003). An analogous
situation concerns the primordial radioactive nuclide $^{40}$K ($t_{1/2} = 1.28 \cdot 10^9$ yr). Its activity in the samples from the Morteratsch and Forni glaciers ($810 \pm 55$, $770 \pm 200$ Bq kg$^{-1}$) is of the same order of magnitude of the average value for UCC

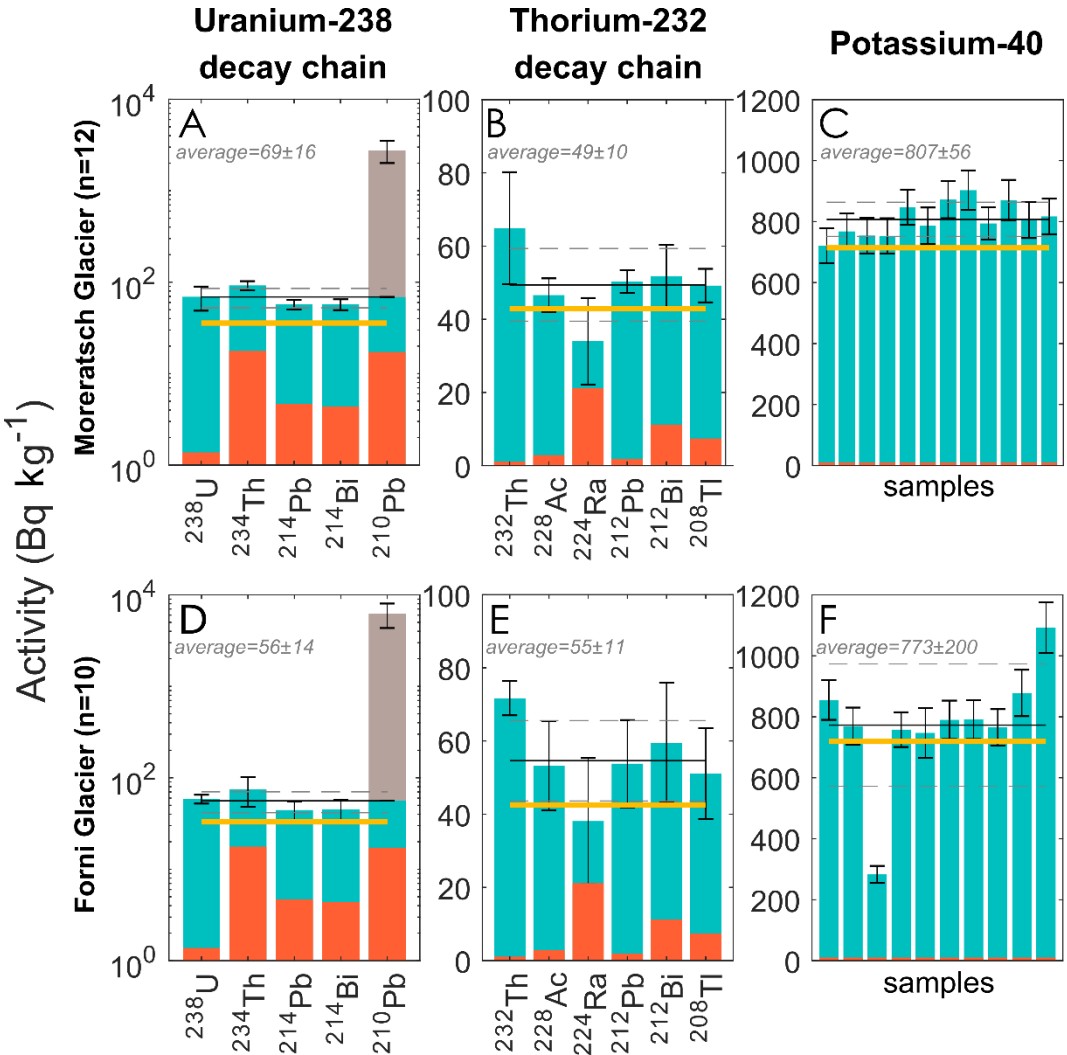

**Fig. 3** Activity of the radionuclides belonging to the decay chains of $^{238}$U and $^{232}$Th and of $^{40}$K. The upper row (panels A-C) refers to the cryoconite samples from the Morteratsch glacier, the lower ones (panels D-F) to the samples collected on the Forni glacier. Red bars represent detection limits, and green bars measured activities. The activity of $^{210}$Pb was divided into supported (green bar) and unsupported fractions (grey bar), considering the upper $^{238}$U decay chain as reference for the supported fraction. Solid (black) and dashed lines correspond to average and standard deviation activity of the decay chains respectively (not considering $^{210}$Pb) and of $^{40}$K (in this case single sample data are shown), and yellow lines to average upper continental crust activity of $^{238}$U, $^{232}$Th and $^{40}$K, gathered from the average UCC elemental concentrations reported by Rudnick & Gao (2003).

concentration of K, i.e. 720 Bq kg$^{-1}$, calculated from the UCC reference (Rudnick and Gao, 2003). This points to a crustal origin for the natural lithogenic radionuclides measured in cryoconite and to the absence of accumulation and/or dilution processes. An exception to this is $^{210}$Pb, which, although being a decay product of $^{238}$U progeny, shows activity levels two orders of magnitude higher than the other $^{238}$U-chain nuclides. The average activities in the samples from the Morteratsch and Forni glaciers are 2,800 ± 800 and 6,200 ± 1,900 Bq kg$^{-1}$ respectively and are statistically different (Student's t test: p-





value < 0.01; degree of freedom = 20; t-value = 5.9) within the two glaciers. Finding such high ²¹⁰Pb activities in samples collected on the surface of glaciers is not completely unexpected. It is common to observe an excess of ²¹⁰Pb in Earth surface environments, due to its dual source. A fraction of ²¹⁰Pb is present in materials of geologic origin because of the internal decay of ²³⁸U progeny (supported ²¹⁰Pb); a second fraction (unsupported ²¹⁰Pb) is found in samples exposed to the

atmosphere and is attributable to the scavenging by precipitation of atmospheric ²¹⁰Pb, produced from the decay of the gaseous ²²²Rn released into the atmosphere from rocks and soils. Given its relatively long half-life (22.3 yr), precipitated ²¹⁰Pb concentrates in surficial environments, but typically its activity doesn't exceed tens or a few hundreds of Bq kg⁻¹ in matrices strongly influenced by atmospheric deposition and rich in organic matter, for which Pb is particularly affine (Strawn and Spark, 2000). In Fig. 4, cryoconite radioactivity is compared to data from literature concerning other

environmental matrices. With respect to lichens and mosses, which are known to be efficient in accumulating radioactive atmospheric species (Kirchner and Daillant, 2002), cryoconite shows a ²¹⁰Pb activity that is, on average, higher by one order of magnitude. Two hypotheses are made to explain the excess found in cryoconite: 1) the two glaciers considered here are located in areas where the atmospheric deposition of ²¹⁰Pb is enhanced; and 2) cryoconite is more efficient at concentrating atmospherically derived radionuclides than lichens and mosses. At the Morteratsch Glacier a comparison has been made

between cryoconite and samples collected from the surface of the moraines surrounding the glacier (Baccolo et al., 2017). The moraine sediments have had a mean activity of 145 ± 30 Bq kg⁻¹ for unsupported ²¹⁰Pb, while in cryoconite it has exceeded 2,500 Bq kg⁻¹. This evidence rejects the first hypothesis: if an anomaly of atmospheric ²¹⁰Pb deposition was present in the Morteratsch valley, it should impact both the surface of the moraines and the surface of the glacier. Several studies have reported high unsupported ²¹⁰Pb activity in cryoconite from different regions of the Earth (Baccolo et al., 2017; Łokas

et al., 2016, 2018; Tieber et al., 2009), suggesting that high ²¹⁰Pb activity is related to characteristics of cryoconite and to interactions with processes occurring on the surface of glaciers.

**4.2 Anthropogenic radioactivity in cryoconite and other environmental matrices**

In Fig. 4, the comparison between radioactive contamination of worldwide lichens, mosses, soils and sediments from other studies (full information is found in the supplementary material), is extended to all of the radionuclides that were found in

excess in cryoconite in this study. Out of all of them, ²¹⁰Pb is the only natural occurring species while the others are anthropogenic in origin. In descending order of average activity in cryoconite, they are: ¹³⁷Cs ($t_{1/2} = 30.1$ yr), ²³⁹⁺²⁴⁰Pu ($t_{1/2} = 24,110$ and $6,536$ yr respectively), ²⁴¹Am ($t_{1/2} = 432.2$ yr), ²⁰⁷Bi ($t_{1/2} = 31.6$ yr), ²³⁸Pu ($t_{1/2} = 87.7$ yr). These nuclides have been released into the environment as a consequence of nuclear incidents and explosions and have been atmospherically transported and deposited globally. For all radionuclides, the activities measured in cryoconite are always higher than those

of other environmental matrices (see Fig. 4). To find samples with activities comparable to the ones found in cryoconite, it would be necessary to consider sites within the vicinity of nuclear tests or incidents. The mean ratios between the activity levels found in cryoconite and in lichens for ¹³⁷Cs, ²³⁹⁺²⁴⁰Pu, ²⁴¹Am, ²⁰⁷Bi, ²³⁸Pu are 9.5, 58, 39, 35, 7 respectively, and the values are even higher when matrices less efficient in accumulating radioactivity are considered. This supports the

**Fig. 4 Radionuclides presenting anomalously high activities in cryoconite compared to other environmental matrices. Activity in cryoconite (green boxes) is compared to data from literature concerning the contamination other matrices sampled in surficial environments (yellow boxes). The number of samples is shown in the lower part of each plot. Given the number of publications from which the displayed data were sourced, they have been listed individually in the supplementary material. All activities were corrected for decay at June 2017, with the exception of $^{210}$Pb, which, being continuously produced in the atmosphere, did not require any adjustment.**

hypothesis that cryoconite accumulates atmospherically derived artificial radionuclides more efficiently than other matrices, as already suggested by exploring unsupported $^{210}$Pb.



The accumulation capability of cryoconite can be observed not only for the most common artificial radionuclides, such as

$^{137}$Cs and $^{239,240}$Pu, but also for less abundant species, such as $^{241}$Am, $^{207}$Bi and $^{238}$Pu. $^{137}$Cs is among the most common long-lived fission products from $^{235}$U and has been released in the environment due to commercial reactor failures and fission bomb test explosions. The plutonium isotopes 239 and 240 also originate from atmospheric weapon tests and nuclear accidents, but the relative contribution from atmospheric tests is larger, since $^{239}$Pu has been the most common fissile material used in pure fission bombs and for igniting fusion devices. Because of their widespread dispersion, $^{137}$Cs and

$^{239,240}$Pu are the most abundant artificial nuclides found in cryoconite. Their mean activities in the samples from Morteratsch and Forni Glaciers are 2,600 ± 3,800 and 1,900 ± 2,900 Bq kg$^{-1}$ for $^{137}$Cs, 80 ± 75 and 4.9 ± 0.9 Bq kg$^{-1}$ for $^{239,240}$Pu (average activities and standard deviations for the Morteratsch and Forni cryoconite respectively). Less abundant nuclides are present in cryoconite with lower concentration, including $^{241}$Am (30 ± 35 and 4 ± 1.5 Bq kg$^{-1}$), $^{207}$Bi (9 ± 7 and 6 ± 2 Bq kg$^{-1}$) and $^{238}$Pu (2.5 ± 2.5 and 0.22 ± 0.08 Bq kg$^{-1}$). Despite being low, such activities are still significant and among the highest ever

found in the environment with respect to these nuclides. Typical environmental activities usually do not exceed 1 Bq kg$^{-1}$ for $^{241}$Am and $^{207}$Bi, and 0.1 Bq kg$^{-1}$ for $^{238}$Pu (Fig. 4), and their rarity is due to their production mechanisms. The presence of $^{241}$Am in the environment is not primarily related to direct deposition (it is present in nuclear power plant spent fuel); it is mostly produced *in situ*, from the decay of its parent nuclide ($^{241}$Pu, t$_{1/2}$ = 14.3 yr), which has been released into the environment alongside other Pu isotopes. Thanks to $^{241}$Pu decay, the environmental activity of $^{241}$Am globally is increasing

and will peak around year 2100 (Thakur and Ward, 2018). $^{238}$Pu is one of the rarest plutonium isotopes produced by commercial reactors and nuclear explosions, and its diffusion is mostly related to the atmospheric re-entry of satellites powered by pure $^{238}$Pu thermoelectric generators (Łokas et al., 2019) and to a smaller degree by the release from nuclear fuel reprocessing plants into marine environment (Bryan et al., 2008). $^{207}$Bi has been released as a consequence of a few high yield thermonuclear explosion tests (Noshkin et al., 2001) and has rarely been observed within the environment. Finding

easily detectable activities in cryoconite for these rare radionuclides, many of which have been released decades ago, is both surprising and unprecedented. Studies focused on these rare radionuclides usually require the application of pre-concentration and separation procedures, but for cryoconite a direct measure of activity was sufficient. These results highlight the potential of this environmental matrix for radioecological monitoring of glaciated areas.

Looking in detail at the two Alpine glaciers considered here, only the activity of Pu isotopes is significantly different

between the two sites, with higher values found in the samples from the Morteratsch Glacier (Student's t test: 0.01 < p-value < 0.02; degree of freedom = 12; t-score = 2.99 for both $^{238}$Pu and $^{239+240}$Pu). $^{241}$Am is also more abundant in the samples from the Morteratsch Glacier, but not significantly because of the large standard deviation. The ratios between the mean activity of the Morteratsch and Forni samples are 15.9, 11.6 and 6.8 for $^{239+240}$Pu, $^{238}$Pu and $^{241}$Am respectively.



### 4.3 Sources of anthropogenic radioactivity in cryoconite

To infer the potential sources of the radioactivity found in cryoconite from Alpine glaciers, isotopic and activity ratios between Pu and Cs isotopes have been calculated (Fig. 5). The use of such ratios has been used as an efficient tool for estimating the provenance of environmental radioactivity, since specific signatures are associated to different sources (Steinhauser, 2019). The atomic ratio $^{240}$Pu/$^{239}$Pu and activity ratio $^{238}$Pu/$^{239+240}$Pu show that, on average, the plutonium-related radioactivity of Morteratsch and Forni cryoconite is compatible with the worldwide signal from global radioactive

fallout (Fig. 5**a**). The latter reflects the composition of the stratospheric Pu reservoir, primarily established in the '60s as a consequence of atmospheric nuclear weapon testing. On average, more than 99 % of the Pu found in cryoconite from the Morteratsch glacier is from global fallout, while for the Forni glacier the average contribution is 95 %, suggesting a non-negligible influence from the Chernobyl accident (~5 %).

By comparing the $^{240}$Pu/$^{239}$Pu ratio of global fallout and of modern snow deposited in the Alps (Gückel et al., 2017), it is
possible to further discuss the Pu sources in cryoconite. Modern Alpine snow has a slightly higher ratio than global fallout (0.21 vs. 0.18), probably because of the partial influence of re-suspended Chernobyl radioactive fallout, which is more enriched in $^{240}$Pu than global fallout (Ketterer and Szechenyi, 2008). Only two cryoconite samples (one from Forni and one from Morteratsch) show a Pu isotopic composition pointing to the Chernobyl influence. They show a ratio of $0.286 \pm 0.006$ and $0.24 \pm 0.02$ respectively, which is even higher than the that of modern Alpine snow. The occurrence of only two samples
bearing a partial Chernobyl signature can be explained by the presence of fallout particles from Chernobyl nuclear fuel in these specimens. The non-volatile constituents of nuclear fuel, such as Pu, were not scattered into the environment homogenously, as in the case of the more volatile $^{137}$Cs, but as micrometric and highly radioactive particles (Sandalls et al., 1993). The presence of even one of such particles in the two samples could be sufficient to explain the anomalies. The other samples present a global signature fully compatible with global fallout and not with modern Alpine snow. This implies that
Pu accumulated in Alpine cryoconite dates back to when the deposition of radionuclides was dominated by global stratospheric fallout, approximately from 1960 to 1980 (Hirose et al., 2008). The deposition of plutonium which is still occurring on the Alpine snowpack, is too weak to influence the isotopic fingerprint of cryoconite. Pu activity in fresh snow ranges from 0.4 to 11.5 µBq kg$^{-1}$ (Gückel et al., 2017), roughly six orders of magnitude lower than the average activity of cryoconite, which is 41 Bq kg$^{-1}$. This has important implications because it suggests that cryoconite is more influenced by a
historic atmospheric fallout rather than the contemporary one, at least when considering Pu. The only source that can provide to cryoconite FRNs from past atmospheric deposition is ice accumulated within the period of maximum deposition of atmospheric radioactivity, when the latter was dominated by global stratospheric fallout. The presence of $^{207}$Bi in cryoconite also supports this hypothesis. In the northern hemisphere this radionuclide was mostly produced during the explosion of the Tzar thermonuclear device in 1961 in Novaja Zemlya (Aarkrog and Dahlgaard, 1984). A few years after this event the $^{207}$Bi
atmospheric contamination decreased until reaching non-detectable levels (Kim et al., 1997). If a considerable amount of

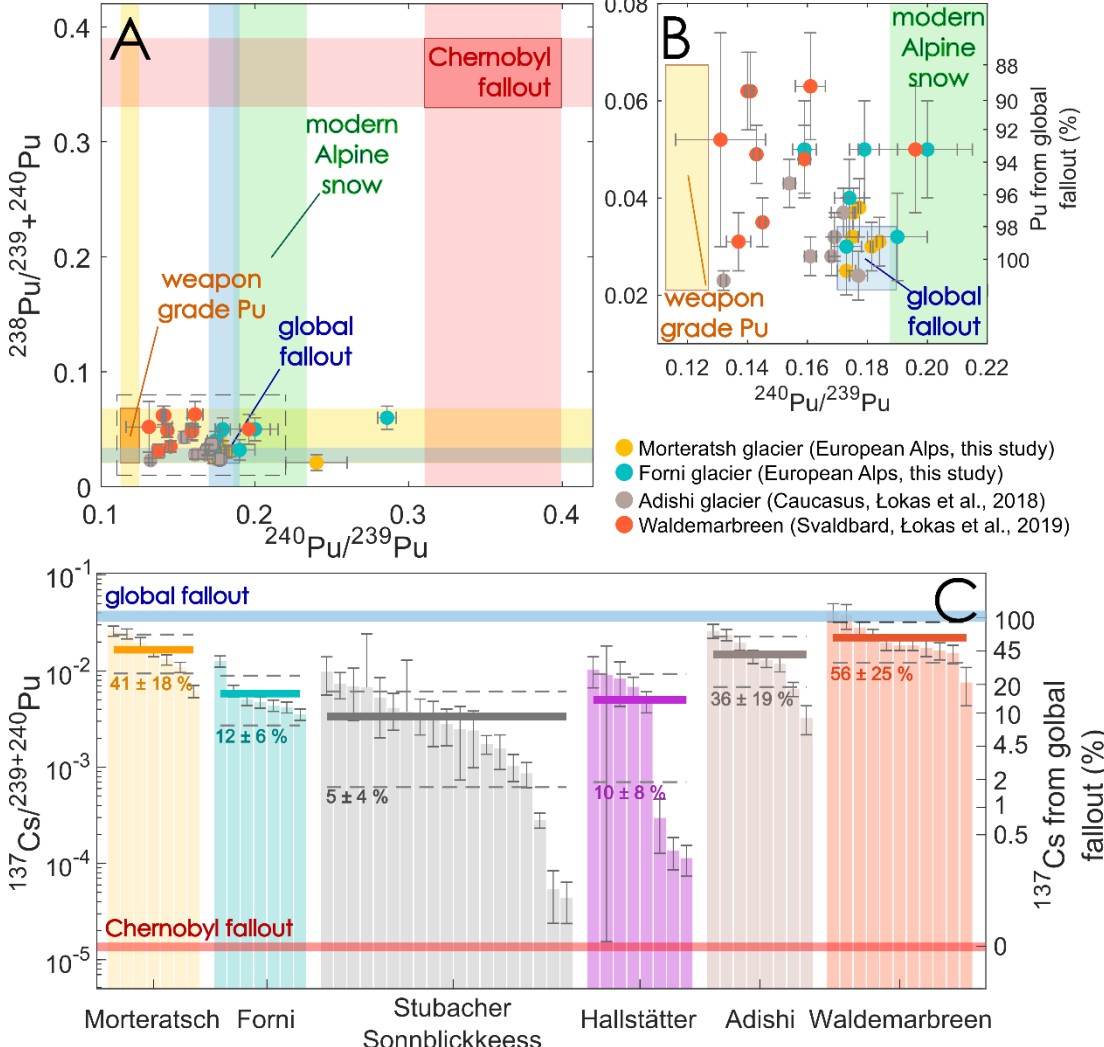

**Fig. 5 Defining the fingerprint of cryoconite radioactivity. Panel A and B: Pu isotopic composition of cryoconite samples (panel B is an enlargement of panel A).** $^{238}$**Pu/**$^{239+240}$**Pu is expressed as an activity ratio, and** $^{240}$**Pu to** $^{239}$**Pu as an atomic ratio. Panel C:** $^{137}$**Cs to** $^{239+240}$**Pu activity ratio of cryoconite. For each glacier the average** $^{137}$**Cs contribution from global fallout (right x-axis) is shown.**

**Coloured and dashed bars represent the average ratios and the standard deviations respectively. In addition to the Forni and Morteratsch glacier samples, data from the Austrian Alps (Stubacher Sonnblickkees and Hallstätter glaciers, Tieber et al., 2009; Wilflinger et al., 2018), Svalbard (Waldemarbreen, Łokas et al., 2019) and from the Caucasus (Adishi glacier, Łokas et al., 2018) are included. Reference ratios (blue bands for global fallout, red bands for Chernobyl fallout, yellow bands for weapon grade Pu, green bands for modern Alpine snow) were taken from literature (Gückel et al., 2017; Ketterer et al., 2008; Cagno et al., 2014;**

**Wilflinger et al., 2018). All values are corrected for decay to June 2017.**

$^{207}$Bi is present in cryoconite, it means that the cryoconite has had the possibility to interact with ice deposited shortly after

1961.





Comparing our results with the data obtained for cryoconite collected in the Caucasus and in regions of the Arctic (Łokas et
al., 2018, 2019), it is possible to see that there are small variations in the radioactive signatures, pointing to secondary local
influences, despite the general features are compatible with global stratospheric fallout (Fig. 5). Caucasian and Arctic
samples are characterized by a lower $^{240}Pu/^{239}Pu$ atomic ratio than the Alpine samples (Fig. 5**a** and **b**). Such a signature is
compatible with the influence of weapon grade Pu, depleted in $^{240}Pu$ (Cagno et al., 2014). It has been argued that samples
from the Caucasus were influenced by the debris spread from the Semipalatinsk Test Site (Kazakhstan), where hundreds of
nuclear test explosions have been carried out (Łokas et al., 2018). The effects of high latitude nuclear polygons (Novaya
Zemlya) and of the reentry of $^{238}Pu$ powered satellites, could explain the non-global fallout contribution observed in the
Arctic cryoconite, which is enriched in both $^{239}Pu$ and $^{238}Pu$ with respect to the Morteratsch and Forni samples (Łokas et al.,
2019). The latter, showing a good agreement with the global fallout reference, rule out the possibility that a fraction of the Pu
found in Alpine cryoconite was produced during the Algerian atmospheric nuclear tests carried out by France between 1960
and 1961.

By studying the ratio between $^{137}Cs$ and $^{239+249}Pu$ activities (Fig. 5**c**), it is possible to infer the potential sources of $^{137}Cs$,
whose activity is by far the highest among the artificial radionuclides found in cryoconite. While global fallout has been
demonstrated as the main source of Pu, the same is not for $^{137}Cs$. On average, the $^{137}Cs$ found in the Morteratsch Glacier
samples associated with global fallout is 41 % with respect to total $^{137}Cs$. For the Forni samples the value is even lower (12
%). The non-global fraction of $^{137}Cs$ found in Alpine cryoconite is attributable to the radioactive contamination released
during the Chernobyl event. The European Alps, and in particular the Eastern Alps, were among the most heavily impacted
areas by Chernobyl fallout, where $^{137}Cs$ was one of the main components (Steinhauser et al., 2014). This is confirmed by the
radioactive signature of cryoconite from two Austrian Alpine glaciers (Eastern Alps) (Tieber et al., 2009; Wilflinger et al.,
2018) whose $^{137}Cs$ content is dominated by Chernobyl contamination (more than 90 %). Samples from the Caucasus also
show a dominant Chernobyl contribution with respect to $^{137}Cs$, while cryoconite from Svalbard is anomalous in being
characterized by a primary influence from global fallout (56 %). This is, however, not unexpected since, among the glaciers
considered in Fig. 5, the Waldemarbreen (Svaldbard) is the farthest from Chernobyl.

While Pu has a dominant global source, $^{137}Cs$ is related both to global and Chernobyl-related fallouts. Pu, together with the
other actinides, is highly non-volatile and its transport mostly takes place through the dispersion of micrometric particles
from nuclear fuel, fission and activation products (Sandalls et al., 1993), while Cs is more volatile and its atmospheric
mobilization during nuclear accidents requires relatively low temperatures. The Pu contamination from Chernobyl was
limited to few hundreds of km from the emission site and could not be efficiently transported for long distances, while $^{137}Cs$
transport was widespread, leaving a strong signature all over Europe (Steinhauser et al., 2014), as also supported by the
radioactive signature of Alpine cryoconite.





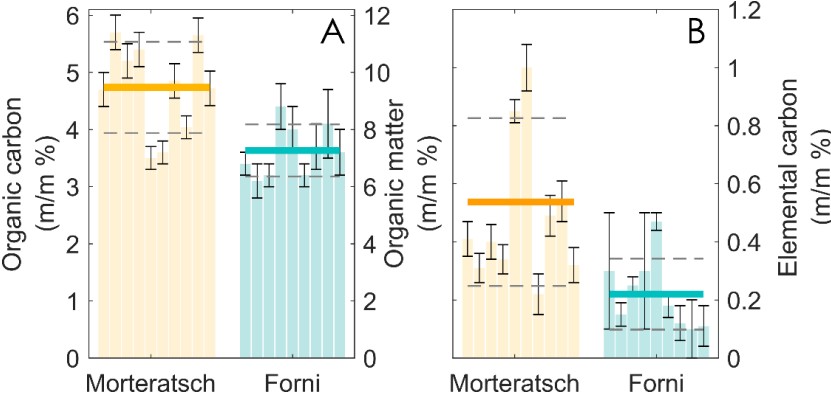


**Fig. 6 Carbonaceous content of cryoconite samples. Panel A refers to organic carbon (and to estimated organic matter), and panel B to elemental carbon. Mean values are depicted alongside standard deviations (coloured and dashed lines).**

## 4.4 Carbonaceous content

Results of carbon analyses are fully presented in Fig. 6. On average (± standard deviation), the carbonaceous composition of the Morteratsch Glacier samples was 4.7 ± 0.7 % m/m in terms of organic carbon and 0.50 ± 0.25% m/m for elemental carbon. Cryoconite from the Forni Glacier contains a lower concentration of both species: 3.6 ± 0.4 % for organic carbon and 0.2 ± 0.2 % for elemental carbon. Organic carbon content has been converted into organic matter content, following the convention by Pribyl (2010). The mean estimated organic matter concentration for the Morteratsch Glacier is 9.4 %, while

for the Forni it is 7.2 %. These values are compatible with data in the wider literature, where organic matter in cryoconite has been reported to vary between 2 and 18 % (Cook et al., 2015).

Very limited information is available about the elemental and/or black carbon composition of cryoconite, despite a great deal of attention having been given to the carbonaceous impurities present in snow in relation to their effect on ice/snow surface darkening (Di Mauro et al., 2017). Our results show that elemental carbon is efficiently accumulated in cryoconite with

respect to Alpine snow, where typical concentrations are at least four orders of magnitude lower (Jenk et al., 2006). Only contaminated urban soils present an elemental carbon concentration comparable to Alpine cryoconite samples ()Lorenz et al., 2006. These findings support the hypotheses by Hodson (2014), who have suggested that cryoconite plays a role in extending the residence time of black and elemental carbon on the surface of glaciers, with implications for the accumulation of hydrophobic contaminants, which have an affinity for these carbonaceous species.

## 375   4.5 Considering radioactivity as a whole

To analyse possible relationships between the different radionuclides, 2-D multidimensional scaling (MDS) and principal component analysis (PCA) have been applied on our data. The first tool has been used to represent the degree of similarity and dissimilarity between the radionuclides (Fig. 7**a**). In the MDS 2-D domain, the radionuclides are grouped within three



clusters which are interpreted as: 1) artificial radionuclides; 2) [238]U-chain nuclides; 3) [232]Th-chain nuclides. Despite [40]K not

belonging to any of these groups, its distance from the [238]U- and [232]Th chain clusters is limited, confirming that K, Th and U

in cryoconite are all associated to its lithogenic component. The most isolated of the nuclides is unsupported [210]Pb, in

accordance to its peculiar biogeochemical cycle.

MDS is able to highlight the different sources of the radionuclides considered in this study: 1) the artificial radionuclides,

whose presence on the glaciers is mostly related to stratospheric fallout; 2) the lithogenic radionuclides which are present in

the mineral fraction of cryoconite; 3) and [210]Pb which is deposited onto the glacier from the lower troposphere by

precipitation. This partitioning is useful for interpreting the differences observed between the two glaciers considered here.

At the Morteratsch Glacier the activity of the stratospherically derived radionuclides (Pu, Am, Bi) is higher than on Forni

Glacier; for [210]Pb the opposite is true (Fig. 4 and **Errore. L'origine riferimento non è stata trovata.**). This pattern may be

related to the altitude of the glaciers. The Morteratsch Glacier basin has a maximum altitude of 4,049 m a.s.l. and an average

elevation higher than 3000 m a.s.l., while the Forni basin is delimited by peaks whose maximum altitude spans from 3200 to

3400 m a.s.l. and only occasionally exceed 3500 m a.s.l.. The lower altitude could explain the higher amount of [210]Pb found

in cryoconite from the Forni Glacier, since the maximum atmospheric scavenging of [210]Pb occurs in the lower troposphere,

below 4,000 m (Guelle et al., 1998). In contrast, the Morteratsch basin, given its high elevation, is more exposed to

stratospheric fallout, perhaps explaining why the cryoconite from this glacier is highly contaminated with Pu isotopes, [241]Am

and [207]Bi.

Results from PCA allow for the distinction of cryoconite sampled from the two glaciers. As seen in Fig. 7**b-c**, the first two

components, mostly the second one, separate the Morteratsch and Forni samples. The nuclides diagnostic for the separation

in PC2 are the anthropogenic ones, which define the negative scores of the Morteratsch samples, and unsupported [210]Pb,

which is linked to the positive scores of the Forni cryoconite. One sample from the Forni Glacier is an outlier with respect to

the others, being characterized by low concentration activity for most of the radionuclides, in particular the artificial ones.

### 4.6 The age of cryoconite and its relationship with ice surface processes

Natural and artificial FRNs are widely used to constrain chronologies in sedimentary environments. Among the nuclides

considered in this work, the most common ones applied for dating are [210]Pb and [137]Cs, while [207]Bi and [241]Am have been

rarely used to mark the period of maximum FRN deposition from atmospheric weapon tests (Appleby, 2008; Kim et al.,

1997). Given the high concentration of radionuclides in cryoconite, it would be interesting to assess if they could be used to

estimate the age of cryoconite itself. The most important issue that makes any attempt at dating challenging, is the complete

absence of a stratigraphic record in cryoconite. In studying cryoconite it is only possible to obtain a set of distinct and

uncorrelated samples. Wilflinger and colleagues (2018) used [210]Pb to infer the mixing age (intended as an approximate mean

age) of cryoconite samples from an Austrian glacier, the Stubacher Sonnblickkees. To attempt the dating, an assumption was





made: once cryoconite is formed, its radioactive content starts decreasing following the decay law. Based on this hypothesis, cryoconite is viewed as a sort of pure concentrated airborne material which is extremely rich in atmospheric derived

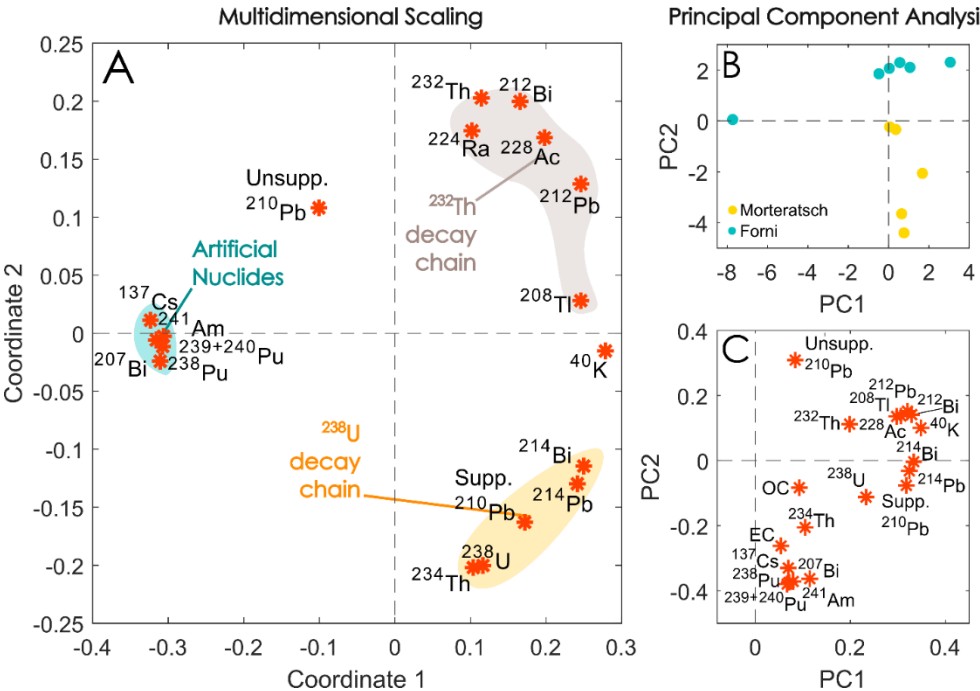

**Fig. 7 Multivariate statistical analysis applied to Alpine cryoconite radioactivity data. Panel A: multidimensional scaling applied to the correlation matrix related to different nuclides. Panel B and C refer to scores and loadings of the first two principal**
**components respectively, calculated through principal component analysis. OC is organic carbon; EC is elemental carbon.**

contaminants, as FRNs. A highly radioactive sample of airborne sediments extracted from fresh snow, was interpreted as a sort of time-zero reference (a primordial cryoconite material), however no further details were given about this specimen. Comparing the $^{210}$Pb activity of cryoconite to the reference, the mixing age of the samples was thus inferred. According to

this conceptual model, older cryoconite presents lower $^{210}$Pb activity in the light of the fact that the more time has passed from its formation, the more profound should be $^{210}$Pb depletion due to the exponential radioactive decay. The estimated ages ranged from a few years to more than a century (Wilflinger et al., 2018). The glacier considered by Wilflinger et al. (2018) is small (less than 1 km$^2$) and is undergoing significant retreat and fragmentation (Kaufmann et al., 2013). The distribution of cryoconite on glaciers is extremely dynamic and is influenced by meteorological processes, local ice

morphology, and supraglacial melting and runoff. It has been observed that within only a few days, single cryoconite holes can form, deepen and collapse (Takeuchi et al., 2018). The transience of surficial glacial environments is also confirmed by glacier moss balls (conglomerations of mineral debris, moss and organic matter forming on the surface of glaciers), whose lifespan was observed not to exceed few years (Hotaling et al., 2019). We thus find it unlikely that a fraction of the



cryoconite sampled on the surface of a small and steep glacier as the Stubacher Sonnblickkees, could form at the end of the
19th century and persist there since then.

We present an alternative hypothesis to link the content of FRNs in cryoconite and its formation age. Our conceptual model arises from an assumption opposite to that of Wilflinger and coauthors (2018): cryoconite is not a static material, its composition changes with time because of the processes taking place on the surface of glaciers. In light of this, the radioactive content of cryoconite is not only subjected to decay, but also to a build-up derived from absorption.
Consequently, the older the cryoconite is, the higher is its $^{210}$Pb content, because it has had a longer time within which to accumulate the radionuclide, which is continuously deposited on the glacier with snow and rain. We hypothesize that the build-up of radioactivity in cryoconite is derived from the interaction between ice, meltwater and cryoconite. During summer ice and snow melt, mobilizing their radionuclide content, including unsupported $^{210}$Pb, which is always present in relatively recent ice (given its lifetime, it is not present at detectable concentrations in ice older than 150-200 years). The interaction
between cryoconite and meltwater containing $^{210}$Pb, explains why the latter is always found at high concentrations in cryoconite, regardless of the geographic context. For artificial FRNs the case is different since they are not continuously deposited on the surface of glaciers; however, they are still present in cryoconite with high activities. Each year during the melting season part of the ice dating back to the peak of atmospheric nuclear tests and to major nuclear incidents, melts out, releasing the artificial nuclides which are transported by meltwater. As for unsupported $^{210}$Pb, cryoconite, which only forms
if meltwater is available (Cook et al., 2015; Takeuchi et al.,2001), retains such nuclides and accumulate a load of artificial radioactivity even if decades have passed since its original deposition on glaciers. The extreme ability of cryoconite is likely related to the affinity of organic matter and the sticky extra-cellular polymeric substances produced by cyanobacteria for radionuclides (Chuang et al., 2015; Gadd 1996).

One observation might corroborate our hypothesis. Wilflinger et al. (2018) reported about high activity of $^7$Be ($t_{1/2}$ = 53 d) in
their samples, of up to 34,000 Bq kg$^{-1}$. $^7$Be is a short-lived cosmogenic radionuclide, deposited from the atmosphere with precipitation. We observed $^7$Be within our samples, but we could not properly quantify it because six months passed between sampling and γ-spectrometry. Finding an excess of $^7$Be in cryoconite, implies that, given its lifetime, the absorption by cryoconite took place in the weeks just before sampling and not when the cryoconite originally formed. The presence of short-lived nuclides suggests that cryoconite continuously accumulates radioactive species through the interaction not only
with meltwater but also with rain, where $^7$Be is always present.

According to our interpretation, cryoconite containing higher concentrations of radionuclides should be older than the cryoconite with lower activities. Beyond this, however, we believe it is difficult to attempt a more precise dating of cryoconite through radioactive decay, even if it remains an interesting task. Too many processes are poorly understood to make a rigorous attempt at present, we first should understand the relationships which exist between the formation of

cryoconite, the geometry of the glacier, the age and displacement of ice, and in particular the exchanges between ice, meltwater and cryoconite.

## 5 Conclusions and future perspectives

We have described the capability of cryoconite to accumulate both artificial and natural FRNs. A comprehensive comparison against other environmental matrices revealed that cryoconite is, excluding samples from nuclear test and incident sites, one

of the most radioactive natural substance found in Earth surface environments, with activities for single radionuclide that can exceed 10,000 Bq kg$^{-1}$, making cryoconite a potentially hazardous material with respect to many legislations.

Our study focused on cryoconite samples from the European Alps but results from other regions of the global cryosphere confirm our findings, proving that the accumulation of radioactivity is not a local phenomenon, but involves worldwide glaciated areas. The capability of cryoconite to accumulate FRNs is so efficient that it has even allowed for a relatively easy

detection of not common FRNs. The high activities detected also made it possible to determine elemental and activity ratios. Cryoconite is thus an extremely promising tool in the fields of radioecology and environmental nuclear forensics.

The use of diagnostic ratios shed light on the sources of radioactivity found in cryoconite. Our analysis revealed that multiple sources, both regional and global, influenced its radioactive signature. Pu related nuclides (Pu and Am isotopes) revealed a dominant source of their presence to be the global stratospheric fallout, associated with atmospheric nuclear tests carried out

in the second half of the 20$^{th}$ century. In contrast, the major contribution for $^{137}$Cs was determined to have come from the 1986 Chernobyl accident. The capability of recording both regional and planetary events, was also suggested by a comparison with literature data concerning cryoconite from other geographic contexts. Some differences were observed in terms of radioactivity signatures and they could be explained considering the impact of local events. It is important to note that currently no information exist about the radioactivity of cryoconite from the Southern Hemisphere. To build a

comprehensive picture of radioactivity in the global cryosphere this is a geographic gap that it would be valuable to close.

There is evidence to suggest that the fundamental process which makes cryoconite a "sponge" for radioactivity, is the interaction between ice and cryoconite itself, through the mediation of meltwater. When glaciers melt, they release and mobilize with meltwater the radionuclides originally preserved in snow and ice layers. Due to the organic matter content and its sticky properties, cryoconite efficiently absorbs the impurities contained in meltwater, in particular those with an affinity

for organic substances, including radionuclides.

This study has focused strictly on the glacial environment, ignoring the fate of cryoconite once it is released by glaciers and transported into the downstream ecosystems. It is likely that owing to meltwater discharge, the radioactivity accumulated in cryoconite is promptly diluted, avoiding any health and ecotoxicological risk. However, caution should be taken considering

those pro-glacial areas in close proximity to the ice, where the dilution could be limited, and some risks could exist. Given
the global relevance of this phenomena, further research should focus on the extra-glacial fate of cryoconite and the contaminants contained within it.

Half a century ago, when nuclear atmospheric testing was a common practice, no one could expect that, thanks to the unique features of glacial environments, the ultimate legacies from those activities would have been maximally concentrated on the surface of glaciers around the world.

## Data availability

Full data are available as supplementary material.

## Author contributions

G.B. conceived the idea of this study, interpreted the data and wrote the manuscript with contributions from all the coauthors; R.A., G.B., B.D.M., A.F., R.S.A. collected the samples; G.B., E.Ł., P.G., M.N. performed the radioactivity
measurements and outlined the potential sources of radioactivity; D.M. and P.P. determined the carbonaceous content of cryoconite; G.B., M.N, M.P. carried out neutron activation analysis; C.C., B.D., V.M., E.P. helped in the interpretation of the data; V.M. handled funding acquisition.

## Competing interests

The authors declare that they have no competing interests.

## Acknowledgments

This study has been supported by the Project of Strategic Interest NextData, funded by the Italian National Research Program PNR 2011-2013, and by the MIAMI (Monitoraggio Inquinamento Atmosferico della Montagna Italiana) project, funded by "Dipartimento per gli affari regionali e le autonomie della Presidenza del Consiglio dei Ministri".

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
