# Peer review of "Cryoconite: an efficient accumulator of radioactive fallout in glacial environments"

_The Cryosphere, 2019_

## Referee Comment (RC1) · Anonymous Referee #1 · 18 Sep 2019

General comments

This study presents high concentrations of radionuclides found in cryoconite collected from two mountain glaciers in European Alps. Cryoconite is organic and inorganic sediment on glacial ice and has been studied chemically and biologically on world-wide glaciers. However, there has been still limited information on radionuclides in cryoconite. The manuscript is well-written and contains interesting analytical results, which were properly discussed in terms of natural and anthropogenic sources of radionuclides. However, I have some concerns on the discussion of the accumulation processes of radionuclides in cryoconite. I would strongly suggest to revise the points indicated below before the publication.

Major points

1. I would strongly suggest to divide the section of "Results and discussion" into two sections: i.e. "Results" and "Discussion", which would present the context of this paper more efficiently.

2. Use carefully the terms of "cryoconite" and "cryoconite granules". "Cryoconite" means bulk sediment on glacier ice, but "cryoconite granules" mean spherical aggregations of the sediment. This difference is particularly important when authors discuss the resident time of substances or elements in cryoconite. In many cases in the text, cryoconite should be replace to the cryoconite granules, please check it throughput the text.

3. High concentrations of 210Pb in the cryoconite is interesting. Authors concluded that it is a result from interaction between ice, meltwater, and cryoconite. However, this could be discussed more carefully with previous works. For example, there has been a quantitative study of accumulation of 210Pb in snow and ice on an alpine glacier in Europe (Gäggeler et al., 1983). The age of ice at the sampling sites in this study seems to be important to explains the high 210Pb concentrations. If available, it would be worth to show the exact locations of samples on the glaciers and age of ice (or estimation based on the glacial ice movement). In terms of role of organic matter or biological activity for 210Pb in cryoconite, there have been many studies on the process of 210Pb (or 210Po) in organics in marine or other environments (e.g. Kim et al., 2012; Fowler et al., 2011) and also on accumulations of heavy metals in snow algal cells (Fjerdingstad, 1973), which would help to understand why the 210Pb was concentrated in cryocontie. Nagatsuka et al. (2010) showed the variations in stable isotopes of Pb in different organic and mineral fractions in cryoconite, which may also be worth to discuss the accumulation process of Pb in cryoconite.

4. The accumulation process of elements in cryoconite should not rely on their radioactivity, but on the chemical or biological properties of each element regardless of radioactive or stable elements. Some statements in the text are misleading. For example, authors say that "cryoconite accumulates radioactivity" in L92, but cryoconite does

not accumulate radioactivity, but may chemically accumulates the elements including radionuclides. Same in many places (e.g. L435-446). Please present it correctly.

5. In conclusion, authors mentioned that cryoconite is a potentially hazardous material, however, I would think that this is an excessive statement and out of the context of this manuscript. There was only one sample that exceeded 10000 Bq kg-1 in this study. Also, the limitation of legislations on the radioactivity in environmental materials should be shown and their potential risk for human health should be quantitatively discussed if authors want to use this statement. I would suggest to state the conclusion more objectively.

Minor comments

1. Title: I would not be sure that cryoconite can be an efficient monitor of radioactive fallout. Use of "monitor" here is very vague. Based on the conclusion, we might detect artificial radioactive elements in cryoconite, but it seems to be difficult to know the time and amounts of their fallout. There might be more proper title for the manuscript.

2. L27 It would be worth to state the specific interaction between cryoconite and the environments.

3. L28 Again, what is "an ideal monitor"? Need specific explanation.

4. L47 Insert "on the ice surface" after "a dispersed material".

5. L53 Insert "granules" after "cryoconite".

6. L63 Insert the year of Meese et al.

7. L82-85 Nagatsuka et al. (2010) could also be worth to be cited here.

8. L92 Please state properly that cryoconite doesn't accumulates radioactivity, but accumulates radioactive elements.

9. L103-116 It would be worth to add more information of the two glaciers in this study,

for example, the reason why the authors selected these two glaciers for this study and difference of mass balance, glacial flow velocity or estimated age of ice at the sampling sites between the glaciers.

10. L127-128 Please show exact locations of the 12 samples for Morteratsch and 10 for Forni Glaciers on the map of Fig.2, and add their coordinates and altitudes in the Table S1. This is important to discuss the resident time of cryoconite on the glaciers. Also, please show the total amounts (dry weight) of cryoconite used in this study.

11. L168 "Pearson's correlation coefficient" instead of "Peason correlation"

12. L179-184 This part, which presents mostly previous works, should not be in Results, but be moved to Introduction or discussion section.

13. L214 Suggest to start a new paragraph here.

14. L228 It is very good to compare the results with those of other environmental samples. But, it would be better to compare with only studies in Europe in order to show whether cryconite accumulates the elements or not. Because the radionuclide activities can vary with geographical locations, i.e. the distance from the source.

15. L254-255 Please clarify that this statement is from previous works (need references) or from this study.

16. L274-278 The difference of the two glaciers is interesting and can be discussed in more detail here or later.

17. L304-305 Specify the time difference between "historic" and "contemporary". It would be worth to mention the age of ice at the sampling site if available.

18. L324-347 and Fig.5 It would be worth to show a map showing the geographical locations of the Caucasus, Svalbard, Chernobyl, Semipalatinsk, and the glaciers of this study.

19. L360-374 Organic carbon contents in cryoconite seems to be significantly different

between the two glaciers. Please explain why.

20. L383-400 It is interesting that the radionuclides differ between the two glaciers. Authors discussed it with only the difference of altitude of the glaciers, but it needs to be discussed more carefully. What is the geology of the bedrock of the two glaciers? There is a significant difference of carbon contents in cryoconite, which could also affect the accumulation of radionuclides? Please discuss also the difference of age of ice of the glaciers.

21. L410 Clarify whether this text means "cryoconite" or "cryoconite granules".

22. L420 Again, insert "granules" after cryoconite, and check it throughout this paragraph.

23. L434 What is "absorption"? Explain and clarify it.

24. L447-449 It is likely, but please explain more carefully how the organics incorporate the elements, by microbial metabolism, or by just chemical combination? Also, discuss it with the difference of organic matter contents between the two glaciers.

25. L456 "older" is very vague. What this "old" exactly means? Does it mean the time from deposition on the glacier, or from the formation of cryoconite granules?

26. L481 Again, "makes cryoconite a "sponge" for radioactivity" is misleading expression. It is not a sponge for radioactivity, but might be a sponge for the elements including the radionuclides.

References

Fjerdingstad, E. (1973). Accumulated concentrations of heavy metals in red snow algae in Greenland. Schweizerische Zeitschrift für Hydrologie, 35(2), 247-251.

Fowler, S. W. (2011). 210Po in the marine environment with emphasis on its behaviour within the biosphere. Journal of environmental radioactivity, 102(5), 448-461.

Gäggeler, H., Von Gunten, H. R., Rössler, E., Oeschger, H., & Schotterer, U. (1983). 210 Pb-Dating of cold alpine firn/ice cores from Colle Gnifetti, Switzerland. Journal of Glaciology, 29(101), 165-177.

Kim, G., Kim, T. H., & Church, T. M. (2012). Po-210 in the environment: biogeochemical cycling and bioavailability. In Handbook of Environmental Isotope Geochemistry (pp. 271-284). Springer, Berlin, Heidelberg.

Nagatsuka, N., Takeuchi, N., Nakano, T., Kokado, E., & Li, Z. (2010). Sr, Nd and Pb stable isotopes of surface dust on Ürümqi glacier No. 1 in western China. Annals of Glaciology, 51(56), 95-105.
* * *

---

## Referee Comment (RC2) · Elizabeth Bagshaw (Referee) · 14 Oct 2019

General comments

This paper is an interesting summary of an emerging research area, that of cryoconite as a record of fallout radionuclides and a potential concentrator of impurities. I cannot comment on the nuclide analysis methods, but they seem sound and reference other publications, so I have confidence in the research team to conduct these analyses appropriately. The paper is generally well written and presents some interesting results. I particularly liked the dating hypothesis discussion, and was gratified that the authors acknowledge that this is an area that needs more work, rather than trying to sew up everything in this one paper. I found the carbon discussion a little distracting and would

recommend removing this section since it didn't really contribute to the main story.

The figures were sometimes a little confusing, with too much colour and too much information presented simultaneously. I make some suggestions for improvement below, but would certainly recommend testing for colour-blind readers as a minimum, and improving/simplifying the labelling and shortening the captions.

I would also suggest that the abstract is rewritten to better reflect the key findings of the paper (which I understand as): that cryoconite is an important concentrator of FRNs; that FRNs in different Alpine Glaciers are similar to each other; that Alpine glaciers are similar to other glaciers but show important differences with respect to proximity to some sources; and that FRNs could be a way of dating cryoconite, since they accumulate over time (in contrast to previous suggestions). As written now, I didn't think it represented the key findings of the paper. The distinction between local and global sources is also confusing, since most cryoconite research considers 'local' to be within catchment (when defining, for example, debris sources or microbial seeding grounds). Instead, perhaps be specific that Chernobyl impacted the Alpine Glaciers but not so much the Svalbard one. The processes description in the abstract is particularly weak and I didn't think very relevant. Use the words for your dating hypothesis instead.

Specific amendments P1, L20: 'extremely rich' is too subjective

L23: 'among the most radioactive environmental matrices' is rather vague – can you be specific?

L27: can you elaborate here? What specific aspects of their interaction?

P2 L33: 'the latter of these' instead 'of which'

Suggest combining the first two paragraphs, they are very short.

L45: 'incoherent' is awkward, suggest replacing with 'unconsolidated'

L46: I would dispute that cryoconite requires abundant meltwater to form - it is found

on ice surfaces in Antarctica with extremely limited quantities of meltwater

L50: please include a reference on the role of cyanobacteria

L53: I think this is specific to cryoconite granules – cryoconite may be present without forming granules (eg. Antarctica). I would suggest adding 'granules' to the end of this sentence.

L59: could you include some example references or a review paper here?

Figure 1: please indicate the scale on A and B, or state the approx. hole diameter in the text

P4 L104: could you include some example references or a review paper here?

L106: can you tell us when it detached, rather than 'few years'?

L114: tell us why this is favourable for the formation of cryoconite (simply put: because there is more source material)

L118: define 'clean' – how were they cleaned? Deionised water? Ethanol? Between samples? In what vessels were the samples stored, and how were they treated?

L128: How were the sampling sites chosen, and how widespread were they?

L131: this is the assumption of all papers. Instead of saying that the material are not published, I would suggest rephrasing to say that accompanying gamma spectroscopy data can be found in the 2017 publication.

L172: Is the equation and description of Pearson Correlation necessary? I think the reference is sufficient, but leave this at the author's discretion

Figure 3: Can the lines be labelled on the plot rather than in the very long caption? For example, the yellow (continental crust), black (average (mean?!)) and dashed (st dev) could be labelled instead, reducing the overlong caption. I would also check the colours for use by colour-blind readers – perhaps patterns could be used instead?

L190: I don't understand why the difference between K40 and the UCC is not significant, but the difference between U and Th is significant, considering the scales on the activity plots. This is because this is beyond my subject area, but may be the case for other readers, so I suggest clearer explanation on the differing scales and assignation of significant differences.

Figure 4: Nice clear plot, although check the colours again.

L246: This is really interesting!

L260: Include a reference

L272: Fascinating!

L362: Include a ref on plutonium deposition in snow here

Figure 5 is quite baffling. I like the labelled sections, but it's unclear whether the labels refer to a whole box or a specific point. The percentage lines on the lower plot are also quite confusing – would this be better presented in a table?

L360-366: include more details on this in the methods section

Section 4.4: is this relevant to the overall story of the paper? There are many studies exploring carbon and black carbon content of cryoconite, particularly in Greenland, and I wonder if these data would be more relevant in another comparative study.

L387: typographical error

Figure 7 is slightly confusing, could only the most important be labelled in C?

L406: yes, this would be really cool! You could refer to the work of Tranter, Fountain or Bagshaw on using chloride to date hydrological age of cryoconite in Antarctica as an example if you wanted to include a comparison.

L 431-445: This hypothesis seems sound and defendable, except the supposition that cryoconite only forms when meltwater is available (L445). I would rephrase this.

L466: give examples of the legislations, or remove this sentence (it's not particularly relevant)

L484: I think that rather than 'absorbs', 'binds' would be a better description, since you seem to show that the EPS sticking the granules together binds up the impurities as well

Final sentence is not strictly relevant and a bit literary.

Data availability are not shown. This must be corrected.

---

## Author Comment (AC1) · 25 Oct 2019

**Replies to the referees**

**Referee #1**

**R1:** This study presents high concentrations of radionuclides found in cryoconite collected from two mountain glaciers in European Alps. Cryoconite is organic and inorganic sediment on glacial ice and has been studied chemically and biologically on worldwide glaciers. However, there has been still limited information on radionuclides in cryoconite. The manuscript is well-written and contains interesting analytical results, which were properly discussed in terms of natural and anthropogenic sources of radionuclides. However, I have some concerns on the discussion of the accumulation processes of radionuclides in cryoconite. I would strongly suggest to revise the points indicated below before the publication.

**Reply:** Thank you very much for the comment, we will do our best to update the manuscript considering your comments which we found appropriate and constructive.

**R1:** I would strongly suggest to divide the section of "Results and discussion" into two sections: i.e. "Results" and "Discussion", which would present the context of this paper more efficiently.

**Reply:** if possible, we would like to maintain the current structure of the paper. We have decided to merge the two parts in order to facilitate the reading to people which is not expert in the field of radioecology. In its current structure, the paper is directly divided into "logical" sections, where data are directly discussed in relation to the glaciological context and to the many comparison that we have presented. We believe that dividing results from conclusions could make more difficult to appreciate the unique radiological features of cryoconite, in particular for people not expert in the field of radioecology.

**R1:** Use carefully the terms of "cryoconite" and "cryoconite granules". "Cryoconite" means bulk sediment on glacier ice, but "cryoconite granules" mean spherical aggregations of the sediment. This difference is particularly important when authors discuss the resident time of substances or elements in cryoconite. In many cases in the text, cryoconite should be replace to the cryoconite granules, please check it throughput the text.

**Reply:** we agree with the reviewer, we have modified the text and in particular the section entitles "The age of cryoconite and its relationship with ice surface processes", where we have tried to better highlight the differences between cryoconite and cryoconite granules in relation to the age issue, adding some passages about this point.

**R1:** High concentrations of 210Pb in the cryoconite is interesting. Authors concluded that it is a result from interaction between ice, meltwater, and cryoconite. However, this could be discussed more carefully with previous works. For example, there has been a quantitative study of accumulation of 210Pb in snow and ice on an alpine glacier in Europe (Gäggeler et al., 1983). The age of ice at the sampling sites in this study seems to be important to explain the high 210Pb concentrations. If available, it would be worth to show the exact locations of samples on the glaciers and age of ice (or estimation based on the glacial ice movement). In terms of role of organic matter or biological activity for 210Pb in cryoconite, there have been many studies on the process of 210Pb (or 210Po) in organics in marine or other environments (e.g. Kim et al., 2012; Fowler et al., 2011) and also on accumulations of heavy metals in snow algal cells (Fjerdingstad, 1973), which would help to understand why the 210Pb was concentrated in cryocontie. Nagatsuka et al. (2010) showed the variations in stable isotopes of Pb in different organic and mineral fractions in cryoconite, which may also be worth to discuss the accumulation process of Pb in cryoconite.

**Reply:** we thank the reviewer for having suggested the mentioned paper that we didn't know. The paper is a pioneering study about the application of 210Pb as a dating tool for ice core samples at a high altitude Alpine glacier. Now 210Pb is routinely applied by the ice core community. Assuming a constant depositional rate, 210Pb is more abundant in surficial (and relatively young) snow and ice, while in the deeper layers of the glaciers (which are normally older) it is less abundant because of the radioactive decay. But this is true when dealing with undisturbed and high elevation cold glaciers only, where no melt occurs and where the ice stratigraphy is preserved and not altered by excessive horizontal ice motion. When we consider temperate glaciers subjected to massive melt, this is no longer true, because 210Pb is mobilized by meltwater and the stratigraphic signal is destroyed. This is discussed by Gäggeler et al. (1983) themselves and by some of the cited references (Shotterer et al., 1977, Ambach et al., 1971). Considering the terminal part of the Morteratsch and Forni glaciers we need to highlight two points: 1) at both glaciers the terminal part is subjected to strong melt during summer and in fact the glaciers are experiencing a strong retreat. No net accumulation is possible there and thus the stratigraphic signal of 210Pb into the ice is completely destroyed by the mass loss and by meltwater percolation. 2) the regions of the glaciers where we have sampled cryoconite are located downstream of two steep icefalls. Such glaciological contexts are known for the effects on ice stratigraphy which is completely disarranged (Goodsell et al., 2002). For these two reasons we believe that the age of the ice where we sampled cryoconite doesn't have a relevant role in determining the amount of 210Pb found in cryoconite. We don't have the possibility to estimate the age of the ice at the sampling points, but we also believe that such information could be hardly obtained, considering that the ice stratigraphy is somewhat disturbed in those regions.

We have now included the suggested references about the role of organic matter and biological activity in the accumulation of radionuclides and about the ability of supraglacial debris in accumulating heavy metals and Pb stable isotopes.

**R1:** The accumulation process of elements in cryoconite should not rely on their radioactivity, but on the chemical or biological properties of each element regardless of radioactive or stable elements. Some statements in the text are misleading. For example, authors say that "cryoconite accumulates radioactivity" in L92, but cryoconite does not accumulate radioactivity, but may chemically accumulates the elements including radionuclides. Same in many places (e.g. L435-446). Please present it correctly.

**Reply:** we agree with the reviewer. We have modified several passages in the text where we have stated that "cryoconite accumulate radioactivity". In most cases we now have changed to "accumulate radionuclides". In addition, we have added a passage to better communicate that cryoconite doesn't accumulate radioactivity because it is affine for it, but because it presents a biogeochemical affinity for some species that are, by the way, radioactive:

"*The extreme ability of cryoconite is likely related to the presence of organic matter and extracellular polymeric substances which are affine for heavy metals, including the radioactive ones (Chuang et al., 2015; Gadd 1996; Fowler et al., 2010; Kim et al., 2011). An additional support for the importance of organic matter in this process is also given by previous studies showing that the organic fraction of cryoconite and snow algae accumulates heavy metals associated to anthropogenic atmospheric emissions, such as stable Pb (Fjerdingstad, 1973, Nagatsuka et al., 2010).*"

**R1:** In conclusion, authors mentioned that cryoconite is a potentially hazardous material, however, I would think that this is an excessive statement and out of the context of this manuscript. There was only one sample that exceeded 10000 Bq kg-1 in this study. Also, the limitation of legislations on the radioactivity in environmental materials should be shown and their potential risk for human health should be quantitatively discussed if authors want to use this statement. I would suggest to state the conclusion more objectively.

**Reply:** a comment from the second referee is very similar to this. We have thus decided to remove this passage from the Conclusions.

**R1:** Title: I would not be sure that cryoconite can be an efficient monitor of radioactive fallout. Use of "monitor" here is very vague. Based on the conclusion, we might detect artificial radioactive elements in cryoconite, but it seems to be difficult to know the time and amounts of their fallout. There might be more proper title for the manuscript.

**Reply:** we agree with the reviewer. The new proposed title is *"Cryoconite: an efficient accumulator of radioactive fallout in glacial environments"*

**R1:** L27 It would be worth to state the specific interaction between cryoconite and the environments.

**Reply:** we have added a sentence in the abstract to better present what we mean: *"We hypothesize that the impurities originally preserved into ice and mobilized with meltwater during summer, including radionuclides, are accumulated in cryoconite because of their affinity for organic matter, which is abundant in cryoconite."*

**R1:** L28 Again, what is "an ideal monitor"? Need specific explanation.

**Reply:** we have now changed to "*matrix*"

**R1:** L47 Insert "on the ice surface" after "a dispersed material".

**Reply:** done.

**R1:** L53 Insert "granules" after "cryoconite".

**Reply:** done.

**R1:** L63 Insert the year of Meese et al.

**Reply:** done.

**R1:** L82-85 Nagatsuka et al. (2010) could also be worth to be cited here.

**Reply:** we have added the reference.

**R1:** L92 Please state properly that cryoconite doesn't accumulates radioactivity, but accumulates radioactive elements.

**Reply:** we have changed accordingly.

**R1:** L103-116 It would be worth to add more information of the two glaciers in this study, for example, the reason why the authors selected these two glaciers for this study and difference of mass balance, glacial flow velocity or estimated age of ice at the sampling sites between the glaciers.

**Reply:** As partially explained in a previous reply, we don't have the possibility to estimate the age of the ice where we have sampled cryoconite. The two glaciers present similar features: they are both experiencing a strong retreat as a consequence of a notably negative mass balance. We have selected them because of their size, we

know from our experience that cryoconite is more abundant in larger glaciers than in small ones. In addition, it should be mentioned that both the glaciers are relatively easy to access and are thus suited for multiple expeditions during summer, so as to construct multi-year records in the next years. We have included some passages to take into account these points.

**R1:** L127-128 Please show exact locations of the 12 samples for Morteratsch and 10 for Forni Glaciers on the map of Fig.2, and add their coordinates and altitudes in the Table S1. This is important to discuss the resident time of cryoconite on the glaciers. Also, please show the total amounts (dry weight) of cryoconite used in this study.

**Reply:** unfortunately we don't have the exact information about sample position and elevation. The regions of the glaciers where we sampled cryoconite are extremely dynamic. They are at same time experiencing a strong retreat (tens of meters per year) and subjected to ice horizontal flow (10-20 m per year, an accurate description of the Morteratsch glacier can be found in Rossini et al. (2018), for the Forni glacier we have available the information from the yearly monitoring activity carried out by the regional glaciological service). Considering the continuous evolution of the glacier surface in the sampling area we believe that knowing the exact position of the samples (which continuously changes) is not of primary importance, for this reason we have decided to highlight the region of the glaciers where we sampled cryoconite and not their exact location. We have decided not to add a table with the punctual details about samples, but we have added some passages in the section dedicated to site description, where we specify the altitude at which we have collected the samples and their mass.

**R1:** L168 "Pearson's correlation coefficient" instead of "Peason correlation"

**Reply:** adjusted.

**R1:** L179-184 This part, which presents mostly previous works, should not be in Results, but be moved to Introduction or discussion section.

**Reply:** we have now removed the passage which is already present in the introduction.

**R1:** L214 Suggest to start a new paragraph here.

**Reply:** done.

**R1:** It is very good to compare the results with those of other environmental samples. But, it would be better to compare with only studies in Europe in order to show whether cryconite accumulates the elements or not. Because the radionuclide activities can vary with geographical locations, i.e. the distance from the source.

**Reply:** we have considered worldwide data because our intention is to show a global comparison. It is true that for specific nuclear accidents (for example the Chornobyl one) the distance from the source is important. But this is not the case for other sources, such as the stratospheric fallout related to atmospheric nuclear tests. In this case the fallout is not coming from a specific source, but from a diffused source, making impossible to estimate a distance between the sampling site and radioactive source. The primary aim of our comparison is to show that cryoconite, regardless the geographic position, is the most efficient radioactivity accumulator. For this reason, we would prefer to maintain the comparison as it is. In addition, we note that if we considered only European data, we would have few data available and the comparison would be less significant.

**R1:** L254-255 Please clarify that this statement is from previous works (need references) or from this study.

**Reply:** we have modified this passage including references and more information about its novelty

**R1:** L274-278 The difference of the two glaciers is interesting and can be discussed in more detail here or later.

**Reply:** A discussion about the differences observed at the two glaciers in relation to the different organic matter content of cryoconite is now presented in section 4.5:

*"Another factor that should be considered to explain the stronger contamination of cryoconite from the Morteratsch glacier, is the higher concentration of carbonaceous compounds in cryoconite from this glacier, for which radionuclides are particularly affine (Chuang et al., 2015; Gadd 1996; Fowler et al., 2010; Kim et al., 2011)."*

**R1:** L304-305 Specify the time difference between "historic" and "contemporary". It would be worth to mention the age of ice at the sampling site if available.

**Reply:** unfortunately we don't know the age of ice at the sampling sites. To avoid any misunderstanding, we have replaced the term "historic" with past.

**R1:** L324-347 and Fig.5 It would be worth to show a map showing the geographical locations of the Caucasus, Svalbard, Chernobyl, Semipalatinsk, and the glaciers of this study.

**Reply:** we have updated Fig. 1 including a large scale map where the locations cited in the present work are shown.

[Figure]

**R1:** L360-374 Organic carbon contents in cryoconite seems to be significantly different between the two glaciers. Please explain why.

**Reply:** we can make a hypothesis to explain the difference. Cryoconite sampled at the Morteratsch glacier has been obtained from an elevation between 2100 and 2300 m a.s.l., Forni cryoconite has been obtained from a higher elevation, between 2600 and 2800. The significant elevation difference could be an important factor to explain the difference in terms of organic carbon content. A higher elevation implies lower temperatures, a shorter summer season and thus a less pronounced biochemical activity. We have added a passage to present this hypothesis:

*"Cryoconite from the Morteratsch glacier presents a higher concentration of both organic and elemental carbon than the one from the Forni glacier (Student's t test for organic carbon concentration: $0.0010 < p\text{-}value < 0.0025$; degree of freedom = 9; t-score = 3.80; for elemental carbon concentration: $0.01 < p\text{-}value < 0.05$; degree of freedom = 7; t-score = 3.11). We hypothesize that elevation has a role in explaining this difference. Cryoconite from the Morteratsch glacier have been sampled at an elevation between 2100 and 2300 m a.s.l., while samples from the Forni glacier have been collected between 2600 and 2800 m a.s.l. A higher elevation implies lower temperatures, a shorter summer season and thus a less pronounced biochemical activity, which is in accordance with the lower organic carbon content observed in cryoconite at the Forni glacier."*

**R1:** L383-400 It is interesting that the radionuclides differ between the two glaciers. Authors discussed it with only the difference of altitude of the glaciers, but it needs to be discussed more carefully. What is the geology of the bedrock of the two glaciers? There is a significant difference of carbon contents in cryoconite, which could also affect the accumulation of radionuclides? Please discuss also the difference of age of ice of the glaciers.

**Reply:** we agree that the discussion here can be deepened here. We don't think that geology has an important role, despite the two sites actually present different lithologies (regolith for the Morteratsch and schists for the Forni, but the elemental composition is not so different, we have unpublished data about this). Considering the previous comment, we can assume that elevation has two effects on cryoconite radioactivity. The first one is already discussed in the text, the second one is indirect and involves the different concentration of carbonaceous compounds. Cryoconite from the Morteratsch glacier have been sampled at lower altitude than Forni and for this reason it is more abundant in organic matter (see previous comment). Given the affinity for organic matter of many radionuclides this could be an important factor to explain the higher contamination observed at Forni for many nuclides. We have added a passage to take into account this second hypothesis:

*"Another factor that should be considered to explain the stronger contamination of cryoconite from the Morteratsch glacier, is the higher concentration of carbonaceous compounds in cryoconite from this glacier, for which radionuclides are particularly affine (Chuang et al., 2015; Gadd 1996; Fowler et al., 2010; Kim et al., 2011)."*

**R1:** L410 Clarify whether this text means "cryoconite" or "cryoconite graniles".

**Reply:** in the considered work the dating method is used to estimate the age of cryoconite, not of cryoconite granules. In the section dedicated to the links between cryoconite, its age and the glacial environment, we have now tried to clarify the differences between cryoconite, cryoconite granules and the respective age in relation to aggregation and dissolution processes involving cryoconite.

**R1:** L420 Again, insert "granules" after cryoconite, and check it throughout this paragraph.

**Reply:** we would like to maintain cryoconite here. We believe that the accumulation of radionuclides is something that involves cryoconite regardless its aggregation state. In the present paragraphs we have made several changes to better highlight the differences between cryoconite, cryoconite granules and age.

**R1:** L434 What is "absorption"? Explain and clarify it.

**Reply:** we agree that this is not the best term. We have thus decided to change it with "continuous accumulation".

**R1:** L447-449 It is likely, but please explain more carefully how the organics incorporate the elements, by microbial metabolism, or by just chemical combination? Also, discuss it with the difference of organic matter contents between the two glaciers.

**Reply:** we cannot go too much in detail here because we don't have direct observations about the binding mechanism between radionuclides, organic matter and/or microbes. We have reviewed the literature about this topic which mostly focuses on the marine environment. It seems that radionuclides are preferentially bound to organic colloid and polymeric extracellular macromolecules rather than being incorporated into microbial cells (Gadd 1996; Yang et al., 2013; Chuang et al., 2015). We have already added a section where we discuss the differences observed at the two glaciers in relation to the different content of organic matter in cryoconite. We have now changed the passage as follows:

*"The extreme ability of cryoconite is likely related to the presence of organic matter and extracellular polymeric substances which are affine for heavy metals, including the radioactive ones (Chuang et al., 2015; Gadd 1996; Fowler et al., 2010; Kim et al., 2011).*

**R1:** L456 "older" is very vague. What this "old" exactly means? Does it mean the time from deposition on the glacier, or from the formation of cryoconite granules?

**Reply:** cryoconite formation is a complicated process which involves biotic and abiotic mechanisms that are only partially understood. When we talk about "old" cryoconite, without referring to granules, we mean cryoconite that formed on the glacier long time ago, regardless of its aggregation state. Cryoconite is composed by local material (englacial debris, local sediments from the moraines), by remote material (aeolian dust) and by allochthonous matter (organic matter formed at the glacier surface). Cryoconite formation is definitely a complex process that is not yet fully understood. Within this context it is important to distinguish between cryoconite and cryoconite granules, as the reviewers has correctly highlighted. A cryoconite granule can be of recent formation, but the cryoconite matter can be significantly older. In the present section when we talk about cryoconite age, we refer to the time passed since its original formation as a consequence of the processes cited above, regardless of its aggregation state. We have made many changes to this section and we hope that these points are now sufficiently clear.

**R1:** L481 Again, "makes cryoconite a "sponge" for radioactivity" is misleading expression. It is not a sponge for radioactivity, but might be a sponge for the elements including the radionuclides.

**Reply:** we have now changed: *"There is evidence to suggest that the fundamental process which makes cryoconite a "sponge" for specific impurities, including radionuclides, is the interaction between ice and cryoconite itself, through the mediation of meltwater."*

---

## Author Comment (AC2) · 25 Oct 2019

**Referee #2**

**R2:** General comments. This paper is an interesting summary of an emerging research area, that of cryoconite as a record of fallout radionuclides and a potential concentrator of impurities. I cannot comment on the nuclide analysis methods, but they seem sound and reference other publications, so I have confidence in the research team to conduct these analyses appropriately. The paper is generally well written and presents some interesting results. I particularly liked the dating hypothesis discussion, and was gratified that the authors acknowledge that this is an area that needs more work, rather than trying to sew up everything in this one paper. I found the carbon discussion a little distracting and would recommend removing this section since it didn't really contribute to the main story.

**Reply:** Thank you very much for the positive comment. Regarding the section dedicated to carbonaceous matter, we would prefer to maintain it. We agree with the reviewer that in the first draft the section have sounded a little bit disconnected from the other sections of the manuscript. Reviewer 1 has asked for a deeper discussion about the relationships between cryoconite radioactivity and organic matter, also in relation to the differences observed between the two glaciers. We have expanded these sections and for this reason we would like to maintain it.

**R2:** The figures were sometimes a little confusing, with too much colour and too much information presented simultaneously. I make some suggestions for improvement below, but would certainly recommend testing for colour-blind readers as a minimum, and improving/simplifying the labelling and shortening the captions.

**Reply:** we have modified many of the figures, enhancing the color contrast, using markers in place of colors and trying to simplify them.

**R2:** I would also suggest that the abstract is rewritten to better reflect the key findings of the paper (which I understand as): that cryoconite is an important concentrator of FRNs; that FRNs in different Alpine Glaciers are similar to each other; that Alpine glaciers are similar to other glaciers but show important differences with respect to proximity to some sources; and that FRNs could be a way of dating cryoconite, since they accumulate over time (in contrast to previous suggestions). As written now, I didn't think it represented the key findings of the paper. The distinction between local and global sources is also confusing, since most cryoconite research considers 'local' to be within catchment (when defining, for example, debris sources or microbial seeding grounds). Instead, perhaps be specific that Chernobyl impacted the Alpine Glaciers but not so much the Svalbard one. The processes description in the abstract is particularly weak and I didn't think very relevant. Use the words for your dating hypothesis instead.

**Reply:** We have modified the abstract considering the suggestions from the reviewer. Here the new version:

*"Cryoconite is rich in natural and artificial radioactivity, but a discussion about its ability to accumulate radionuclides is lacking. A characterization of cryoconite from two Alpine glaciers is here presented. Results confirm that cryoconite is significantly more radioactive than the matrices usually adopted for the environmental monitoring of radioactivity, as lichens and mosses, with activity concentrations exceeding 10,000 Bq kg$^{-1}$ for single radionuclides. This makes cryoconite an ideal matrix to investigate the deposition and occurrence of radioactive species in glacial environments. In addition, cryoconite can be used to track environmental radioactivity sources. We have exploited atomic and activity ratios of artificial radionuclides to identify the sources of the anthropogenic radioactivity accumulated in our samples. The signature of cryoconite from different Alpine glaciers is similar and compatible with the stratospheric global fallout and Chernobyl accident products. Differences are found when considering other geographic contexts. A comparison with data from literature shows that Alpine cryoconite is strongly influenced by the Chernobyl fallout, while cryoconite from other regions is more impacted by events such as nuclear test explosions and satellite re-entries. To explain the*

*accumulation of radionuclides in cryoconite, the glacial environment as a whole must be considered, and particularly the interaction between ice, meltwater, cryoconite and atmospheric deposition. We hypothesize that the impurities originally preserved into ice and mobilized with meltwater during summer, including radionuclides, are accumulated in cryoconite because of their affinity for organic matter, which is abundant in cryoconite. In relation to these processes, we have explored the possibility to exploit radioactivity to date cryoconite."*

**R2:** L20: 'extremely rich' is too subjective

**Reply:** we have now removed "extremely".

**R2:** L23: 'among the most radioactive environmental matrices' is rather vague – can you be specific?

**Reply:** we have now changed to *"...cryoconite is significantly more radioactive than the matrices usually adopted for the environmental monitoring of radioactivity, as lichens and mosses..."*

**R2:** L27: can you elaborate here? What specific aspects of their interaction?

**Reply:** we have now added the following passage: "*We hypothesize that the impurities originally preserved into ice and mobilized with meltwater during summer, including radionuclides, are accumulated in cryoconite because of their affinity for organic matter, which is abundant in cryoconite, and in particular for extra-cellular polymeric substances.*"

**R2:** P2 L33: 'the latter of these' instead 'of which'

**Reply:** we have replaced with "cryoconite".

**R2:** Suggest combining the first two paragraphs, they are very short.

**Reply:** we agree with the reviewer, done.

**R2:** L45: 'incoherent' is awkward, suggest replacing with 'unconsolidated'

**Reply:** changed accordingly.

**R2**: L46: I would dispute that cryoconite requires abundant meltwater to form - it is found on ice surfaces in Antarctica with extremely limited quantities of meltwater

**Reply:** we agree, we have now removed "abundant"

**R2:** L50: please include a reference on the role of cyanobacteria

**Reply:** we have now included Langford et al., 2010.

**R2:** L53: I think this is specific to cryoconite granules – cryoconite may be present without forming granules (eg. Antarctica). I would suggest adding 'granules' to the end of this sentence.

**Reply:** done.

**R2:** L59: could you include some example references or a review paper here?

**Reply:** not to increase too much the number of references, we have added a quote to suited papers that had already been cited in the paper.

**R2:** Figure 1: please indicate the scale on A and B, or state the approx. hole diameter in the text

**Reply:** we have now included the scale in the two panels.

**R2:** P4 L104: could you include some example references or a review paper here?

**Reply:** not to increase too much the number of references, we have added a quote to suited papers that had already been cited in the paper.

**R2:** L106: can you tell us when it detached, rather than 'few years'?

**Reply:** of course, it was 2015.

**R2:** L114: tell us why this is favourable for the formation of cryoconite (simply put: because there is more source material)

**Reply:** we have added the passage.

**R2:** L118: define 'clean' – how were they cleaned? Deionised water? Ethanol? Between samples? In what vessels were the samples stored, and how were they treated?

**Reply:** we have added the details. We used sterile disposable pipettes, sterile plastic tubes and spoons cleaned with ethanol between samples.

**R2:** L128: How were the sampling sites chosen, and how widespread were they?

**Reply:** they were extremely common, on average the distance between two adjacent deposits or holes was of few meters. We have chosen the ones where cryoconite was more abundant, we have now added this detail to the text: *"We selected the most abundant cryoconite deposits, so as to have material available for other analyses also: twelve samples have been gathered on the Morteratsch Glacier (between 2100 and 2300 m a.s.l.) and ten on the Forni Glacier (between 2600 and 2800 m a.s.l.), each one consisting in 10-40 g of wet cryoconite."*

**R2:** L131: this is the assumption of all papers. Instead of saying that the material are not published, I would suggest rephrasing to say that accompanying gamma spectroscopy data can be found in the 2017 publication.

**Reply:** we agree, we have rephrased accordingly: *"Part of the data concerning gamma spectrometry applied to cryoconite from the Morteratsch samples has already been published (Baccolo et al., 2017)."*

**R2:** L172: Is the equation and description of Pearson Correlation necessary? I think the reference is sufficient, but leave this at the author's discretion.

**Reply:** we have shortened the passage where we described the link between r and d, however we would to prefer to maintain the equation since MDS is not usually applied to correlation but to distance.

**R2:** Figure 3: Can the lines be labelled on the plot rather than in the very long caption? For example, the yellow (continental crust), black (average (mean?!)) and dashed (st dev) could be labelled instead, reducing the overlong caption. I would also check the colours for use by colour-blind readers – perhaps patterns could be used instead?

Reply: thanks for the suggestion, we have modified the couple figure-caption as it follows, adding a legend and enhancing color contrast:

[Figure]

**Fig. 3 Activity of the radionuclides belonging to the decay chains of $^{238}$U and $^{232}$Th and of $^{40}$K. The upper row (panels A-C) refers to the cryoconite samples from the Morteratsch glacier, the lower ones (panels D-F) to the samples collected on the Forni glacier. Red bars represent detection limits, green bars measured activities. The activity of $^{210}$Pb is divided into supported (green bar) and unsupported fractions (grey bar), considering the upper $^{238}$U decay chain as reference for the supported fraction. Crustal references are inferred from Rudnick & Gao (2003).**

**R2:** L190: I don't understand why the difference between K40 and the UCC is not significant, but the difference between U and Th is significant, considering the scales on the activity plots. This is because this is beyond my subject area, but may be the case for other readers, so I suggest clearer explanation on the differing scales and assignation of significant differences.

**Reply:** potassium is one of the most abundant elements in Earth crust and is also the largest responsible for Earth crust radioactivity. It is a common element found in heaps of different minerals. On the contrary both Th and U are orders of magnitude less abundant and are considered trace elements. In addition, they share peculiar chemical properties, with large ionic radii and high ionic charges. Such features make the two elements incompatible with many common minerals and enhance their fractionation in rare and ancillary minerals which are found in specific lithologic contexts. For this reason, it is not surprising that dealing with K we have found concentrations similar to the Earth crust, while this is not the case for Th and U which are enriched in our samples. In Baccolo et al., 2017 we have discussed the fractionation of some elements in cryoconite. We found that elements usually associated to heavy minerals (as Th and U) are enriched in cryoconite because meltwater flow preferentially removes light minerals and enriches the heavy ones. We have added a passage to explain this point:

*"These values, as seen in Fig. 3, are slightly higher than the average $^{238}U$ and $^{232}Th$ radioactivity of upper continental crust (UCC) reference (Rudnick and Gao, 2003), which is 34 and 43 Bq kg$^{-1}$ for $^{238}U$ and $^{232}Th$ respectively. The difference is probably related to the accumulation in cryoconite of heavy minerals, where both U and Th are typically enriched, because of hydraulic sorting related to meltwater flow (Baccolo et al. 2017)."*

R2: Figure 4: Nice clear plot, although check the colours again.

**Reply:** thanks, we increased the color contrast between cryoconite and not-cryoconite samples, now the graph is:

[Figure]

R2: L246: This is really interesting!

**Reply:** thank you!

R2: L260: Include a reference

**Reply:** we have now added two references, Bossew et al., 2006 and Shabana & Al-Shammari, 2001.

R2: L362: Include a ref on plutonium deposition in snow here

**Reply:** we couldn't find the right position, maybe there's a typo with the line number.

**R2:** Figure 5 is quite baffling. I like the labelled sections, but it's unclear whether the labels refer to a whole box or a specific point. The percentage lines on the lower plot are also quite confusing – would this be better presented in a table?

**Reply:** thank you for the comment, we have exaggerated with colors.... We have better highlighted the boxes, removing the colored bands which could be somewhat confusing. In addition, we changed sample markers to help distinguishing them regardless their color. In the lower graph we have removed the labels about the percentage for each glacier, but we have decided to keep the lines corresponding to average values calculated for each glacier. We agree with the reviewer, to increase readability we have now included a table presenting the average fractions of Pu and 137Cs related to global fallout at the different glaciers. This is the new version of the figure:

[Figure]

**R2:** L360-366: include more details on this in the methods section

**Reply:** we have moved the technical details in the methods section. A full technical description of this procedure can be found in the cited literature, for this reason we have decided no to add further information here.

**R2:** Section 4.4: is this relevant to the overall story of the paper? There are many studies exploring carbon and black carbon content of cryoconite, particularly in Greenland, and I wonder if these data would be more relevant in another comparative study.

**Reply:** we have decided to maintain the section. The first reviewer suggested to better discuss the relationships existing between radionuclides and organic matter content, in particular to explain the differences observed at the two glaciers (cryoconite from Morteratsch is richer in cryoconite than Forni).

**R2:** L387: typographical error

**Reply:** corrected.

**R2:** Figure 7 is slightly confusing, could only the most important be labelled in C?

**Reply:** we agree, there are a lot of labels, but we don't know hot to select only some of them. The only reasonable way would be to remove all the labels, but without labels the figure would be completely unreadable.

**R2:** L406: yes, this would be really cool! You could refer to the work of Tranter, Fountain or Bagshaw on using chloride to date hydrological age of cryoconite in Antarctica as an example if you wanted to include a comparison.

**Reply:** thanks for the advice. We have deepened the discussion about the influence of supra-glacial dynamics on the age of cryoconite. We have added the suggestion references few lines below the point suggested by the reviewer. The new extended passages now read:

*"The distribution of cryoconite on glaciers is extremely dynamic and is influenced by meteorological processes, local ice morphology, and supraglacial melting and runoff. It has been observed that within only a few days, single cryoconite holes can form, deepen and collapse, scattering cryoconite granules downstream on the glacier (Takeuchi et al., 2018). In addition, it is known that cryoconite is far from being a static sediment: cryoconite granules are in fact subjected to uninterrupted changes, such as aggregation and break-up, and their lifetime on glaciers don't exceed a few years (Takeuchi et al., 2010). In Antarctica, where cryoconite holes are usually covered by a permanent ice lid and supra-glacial hydrology is poor, the isolation age (i.e. the time period during which a single cryoconite hole have remained isolated from glacial hydrology) of single cryoconite holes has been estimated through a biogeochemical method: it never exceeds a few years (Fountain et al., 2004; Bagshaw et al., 2007). The transience of surficial glacial environments is also confirmed by glacier moss balls, whose lifespan was observed not to exceed few years (Hotaling et al., 2019). Given these evidences, we find it unlikely that a fraction of the cryoconite sampled on the surface of a small and steep glacier as the Stubacher Sonnblickkees, could form at the end of the 19$^{th}$ century and persist there since then without being subjected to significant compositional changes."*

**R2:** L 431-445: This hypothesis seems sound and defendable, except the supposition that cryoconite only forms when meltwater is available (L445). I would rephrase this.

**Reply:** we have removed that passage

**R2:** L466: give examples of the legislations, or remove this sentence (it's not particularly relevant)

**Reply:** we have removed the passage

**R2:** L484: I think that rather than 'absorbs', 'binds' would be a better description, since you seem to show that the EPS sticking the granules together binds up the impurities as well

**Reply:** we agree, we have now modified with *"binds and accumulates"*

**R2:** Final sentence is not strictly relevant and a bit literary.

**Reply:** we have removed the passage

**R2:** Data availability are not shown. This must be corrected.

**Reply**: we have now included the availability statement: *"Full data are available as supplementary material."*